# Transcriptional profiling at whole population and single cell levels reveals somatosensory neuron molecular diversity

Isaac M Chiu[1,2,6]*, Lee B Barrett[1,2], Erika K Williams[3], David E Strochlic[3], Seungkyu Lee[1,2], Andy D Weyer[4], Shan Lou[5], Gregory S Bryman[1,2], David P Roberson[1,2], Nader Ghasemlou[1,2], Cara Piccoli[1,2], Ezgi Ahat[1,2], Victor Wang[1,2], Enrique J Cobos[1,2,7], Cheryl L Stucky[4], Qiufu Ma[5], Stephen D Liberles[3], Clifford J Woolf[1,2]*

[1]F.M. Kirby Neurobiology Center, Boston Children's Hospital, Boston, United States; [2]Department of Neurobiology, Harvard Medical School, Boston, United States; [3]Department of Cell Biology, Harvard Medical School, Boston, United States; [4]Department of Cell Biology, Neurobiology and Anatomy, Medical College of Wisconsin, Milwaukee, United States; [5]Dana-Farber Cancer Institute, Harvard Medical School, Boston, United States; [6]Department of Microbiology and Immunobiology, Harvard Medical School, Boston, United States; [7]Department of Pharmacology and Neurosciences Institute, University of Granada, Granada, Spain

*For correspondence: isaac_chiu@hms.harvard.edu (IMC); clifford.woolf@childrens.harvard.edu (CJW)

Competing interests: The authors declare that no competing interests exist.

**Abstract** The somatosensory nervous system is critical for the organism's ability to respond to mechanical, thermal, and nociceptive stimuli. Somatosensory neurons are functionally and anatomically diverse but their molecular profiles are not well-defined. Here, we used transcriptional profiling to analyze the detailed molecular signatures of dorsal root ganglion (DRG) sensory neurons. We used two mouse reporter lines and surface IB4 labeling to purify three major non-overlapping classes of neurons: 1) IB4+SNS-Cre/TdTomato+, 2) IB4−SNS-Cre/TdTomato+, and 3) Parv-Cre/TdTomato+ cells, encompassing the majority of nociceptive, pruriceptive, and proprioceptive neurons. These neurons displayed distinct expression patterns of ion channels, transcription factors, and GPCRs. Highly parallel qRT-PCR analysis of 334 single neurons selected by membership of the three populations demonstrated further diversity, with unbiased clustering analysis identifying six distinct subgroups. These data significantly increase our knowledge of the molecular identities of known DRG populations and uncover potentially novel subsets, revealing the complexity and diversity of those neurons underlying somatosensation.

## Introduction

The somatosensory nervous system comprises diverse neuronal subsets with distinct conduction properties and peripheral and central innervation patterns, including small-diameter, unmyelinated C-fibers, thinly myelinated Aδ-fibers, and large-diameter, thickly myelinated Aα/β-fibers (*Basbaum et al., 2009*; *Abraira and Ginty, 2013*). Distinct sets of somatosensory neurons are thought to mediate different functional modalities, such as tactile sensation, proprioception, pruriception and nociception. During development, precise expression of neurotrophic receptors and transcription factors at different times controls the differentiation and connectivity of these diverse sensory afferent populations (*Marmigere and Ernfors, 2007*; *Abraira and Ginty, 2013*). Detection of thermal, mechanical, and chemical stimuli in the external or internal environment by the somatosensory neurons is mediated by expression of specific molecular transducers at their peripheral nerve terminals. For example, transient receptor potential (TRP) ion channels are activated in response to heat, cold, reactive chemicals, leading to

**eLife digest** In the nervous system, a network of specialized neurons—known as the somatosensory system—carries information about sensations including touch, muscle position, temperature and pain. Distinct sets of somatosensory neurons are thought to carry information about the different types of sensations. In young animals, the precise switching on, or 'expression', of genes controls the formation of the network of neurons. However, it is not known exactly which genes are expressed in what types of neurons, where, or when.

Here, Chiu et al. used a technique called flow cytometry using different fluorescent markers to isolate a group of cells called Dorsal Root Ganglion (DRG) neurons in mice. These neurons have long thread-like fibers that extend from the spinal cord to the skin, muscles and joints all over the body. These fibers carry sensory information to the spinal cord, where it can be relayed to the brain and processed. The experiments compared three distinct types of DRG neuron and found that they differed in their ability to send information to other cells.

Chiu et al. analyzed the expression of all the genes in the three types of DRG neurons. Each type of neuron had distinct groups of genes that were being expressed. Also, several genes that are known to be important for sensation were expressed at different levels in the different types of cells. Next, large numbers of single cells were analyzed to find out the finer details about the three types of neuron. These findings made it possible to further divide the DRG neurons into six distinct subsets that matched previously known groups of somatosensory neurons, and also identified new ones.

Chiu et al.'s findings reveal the complexity and diversity of the neurons involved in carrying information about sensations towards the brain. This is an important step in classifying the nervous system, and uncovers many genes previously not linked to sensation. The next challenges lie in understanding how the expression of these genes in each type of neuron relates to their unique roles.

cation influx and action potential generation (*Basbaum et al., 2009*; *Dib-Hajj et al., 2010*; *Dubin and Patapoutian, 2010*; *Julius, 2013*). Given the high degree of cellular diversity of the somatosensory system defined at developmental, anatomical, and functional levels, a classification scheme of different somatosensory neuron subtypes based on the comprehensive set of genes they express is so far lacking. Determining the detailed molecular organization of specific somatosensory neuron subtypes is however necessary for our understanding of their specification, normal function and contribution to disease.

Cell-type specific transcriptome analysis is increasingly recognized as important for the molecular classification of neuronal populations in the brain and spinal cord (*Okaty et al., 2011*). Fluorescence activated cell sorting (FACS) and other neuron purification strategies coupled with transcriptional profiling by microarray analysis or RNA sequencing has allowed detailed molecular characterization of discrete populations of mouse forebrain neurons (*Sugino et al., 2006*), striatal projection neurons (*Lobo et al., 2006*), serotonergic neurons (*Wylie et al., 2010*), corticospinal motor neurons (*Arlotta et al., 2005*), callosal projection neurons (*Molyneaux et al., 2009*), proprioceptor lineage neurons (*Lee et al., 2012*), and electrophysiologically distinct neocortical populations (*Okaty et al., 2009*). These data have uncovered novel molecular insights into neuronal function. Transcriptional profiling technology at the single cell level is transforming our understanding of the organization of tumor cell populations and cellular responses in the immune system (*Patel et al., 2014*; *Shalek et al., 2014*), and has begun to be applied to neuronal populations (*Citri et al., 2012*; *Mizeracka et al., 2013*). This technology has been proposed as a useful approach to begin mapping cell diversity in the mammalian CNS (*Wichterle et al., 2013*).

To begin to define the molecular organization of the somatosensory system, we have performed cell-type specific transcriptional profiling of dorsal root ganglion (DRG) neurons at both whole population and single cell levels. Using two reporter mice, SNS-Cre/TdTomato and Parv-Cre/TdTomato, together with surface Isolectin B4-FITC staining, we identify three major, non-overlapping populations of DRG neurons encompassing almost all C-fibers and many A-fibers. SNS-Cre is a BAC transgenic mouse line expressing Cre under the Scn10a (Nav1.8) promoter (*Agarwal et al., 2004*) which has been

shown to encompass DRG and trigeminal ganglia nociceptor lineage neurons, and in conditional gene ablation studies affects thermosensation, itch, and pain (*Liu et al., 2010*; *Lopes et al., 2012*; *Lou et al., 2013*). A widely used Nav1.8-Cre knock-in mouse line also exists (*Stirling et al., 2005*; *Abrahamsen et al., 2008*), but differs to some extent from the transgenic SNS-Cre mouse line. We find, for example, that SNS-Cre/TdTomato reporter mice label 82% of total DRG neurons, which is slightly greater than Nav1.8-Cre/TdTomato reporter mice (75%) (*Shields et al., 2012*), implying capture of a larger neuronal population. Both the SNS-Cre lineage and Nav1.8-Cre lineage neurons include a large proportion of C-fibers and a smaller population of NF200+ A-fibers (*Shields et al., 2012*). As expected, the majority of TdTomato+ cells (90%) in the SNS-Cre/TdTomato line expressed Scn10a transcript encoding Nav1.8 when tested by RNA in situ *hybridization* (*Liu et al., 2010*). Our second reporter line used Parv-Cre, a knock-in strain expressing Ires-Cre under the control of the Parvalbumin promoter, which has been used in the study of proprioceptive-lineage (large NF200+ A-fiber) neuron function (*Hippenmeyer et al., 2005*; *Niu et al., 2013*; *de Nooij et al., 2013*). Finally we used IB4, which labels the surface of non-peptidergic nociceptive neurons (*Vulchanova et al., 1998*; *Stucky et al., 2002*; *Basbaum et al., 2009*).

Using these mice and the labeling strategies, we were able to FACS purify three major, non-overlapping populations of somatosensory neurons: (1) IB4+SNS-Cre/TdTomato+, (2) IB4−SNS-Cre/TdTomato+, (3) Parv-Cre/TdTomato+ neurons, and analyze their whole transcriptome molecular signatures. Differential expression analysis defined transcriptional hallmarks in each for ion channels, transcription factors and G-protein coupled receptors. Further analysis of hundreds of single DRG neurons identifies distinct somatosensory subsets within the originally purified populations, which were confirmed by RNA in situ hybridization. Our analysis illustrates the enormous heterogeneity and complexity of neurons that mediate peripheral somatosensation, as well as revealing the molecular basis for their functional specialization.

## Results

### Characterization of distinct DRG neuronal subsets for molecular profiling

To perform transcriptional profiling of the mouse somatosensory nervous system, we labeled distinct populations of DRG neurons. We bred SNS-Cre or Parv-Cre mice with the Cre-dependent Rosa26-TdTomato reporter line (*Madisen et al., 2010*). In SNS-Cre/TdTomato and Parv-Cre/TdTomato progeny, robust fluorescence was observed in particular subsets of neurons in lumbar DRG (*Figure 1—figure supplement 1*).

We next analyzed the identity of the SNS-Cre/TdTomato+ and Parv-Cre/TdTomato+ DRG populations by costaining with a set of widely used sensory neuron markers; Isolectin B4 (IB4) (for non-peptidergic nociceptors), Neurofilament-200 kDa (NF200) (for myelinated A-fibers) calcitonin-gene related peptide (CGRP) (for peptidergic nociceptors), and Parvalbumin (for proprioceptors) (*Figure 1A*). IB4 labeled a DRG subset that was completely included within the SNS-Cre/TdTomato population (*Figure 1B*, 98 ± 0.87% IB4+ were SNS-Cre/TdT+; *Figure 1C*, 28.0 ± 1.8% SNS-Cre/TdT+ neurons were IB4+). By contrast, IB4 staining was effectively absent in the Parv-Cre/TdTomato population (*Figure 1B*, 1.18 ± 1.35% IB4+ were Parv-Cre/TdT+). CGRP also fell completely within a subset of the SNS-Cre/TdTomato population and also was absent in the Parv-Cre/TdTomato population (*Figure 1B*, 99.4 ± 0.4% CGRP+ were SNS-Cre/TdT+; 1.5 ± 2.05% CGRP+ were Parv-Cre/TdT+; *Figure 1C*, 45.1 ± 3.9% SNS-Cre/TdT+ were CGRP+). Neurofilament heavy chain 200 kDa (NF200) was expressed by the majority of the Parv-Cre/TdT+ population (*Figure 1B*, 96.1 ± 1.9%), but only a small proportion of the SNS-Cre/TdT+ population (16.9 ± 1.9%). Parvalbumin protein was expressed by the majority of Parv-Cre/TdT+ neurons (*Figure 1C*, 81.4 ± 3.4%), but was absent in the SNS-Cre/TdT+ population (*Figure 1C*, 0.8 ± 0.2%). In the spinal cord, SNS-Cre/TdTomato fibers mostly overlapped with CGRP and IB4 central terminal staining in superficial dorsal horn layers (*Figure 1—figure supplement 1*). By contrast, Parv-Cre/TdTomato fibers extended into deeper dorsal horn laminae, Clark's Nucleus, and the ventral horn (*Figure 1—figure supplement 1*). Taken together, these observations suggest that these two lineage reporter lines labeled two distinct populations of primary sensory afferents and the SNS-Cre/TdTomato population includes several subsets that can be partly delineated by IB4 staining (Venn diagram, *Figure 1D*). By NeuN staining, SNS-Cre/TdTomato labeled 82 ± 3.0% of all DRG neurons, while Parv-Cre/TdTomato

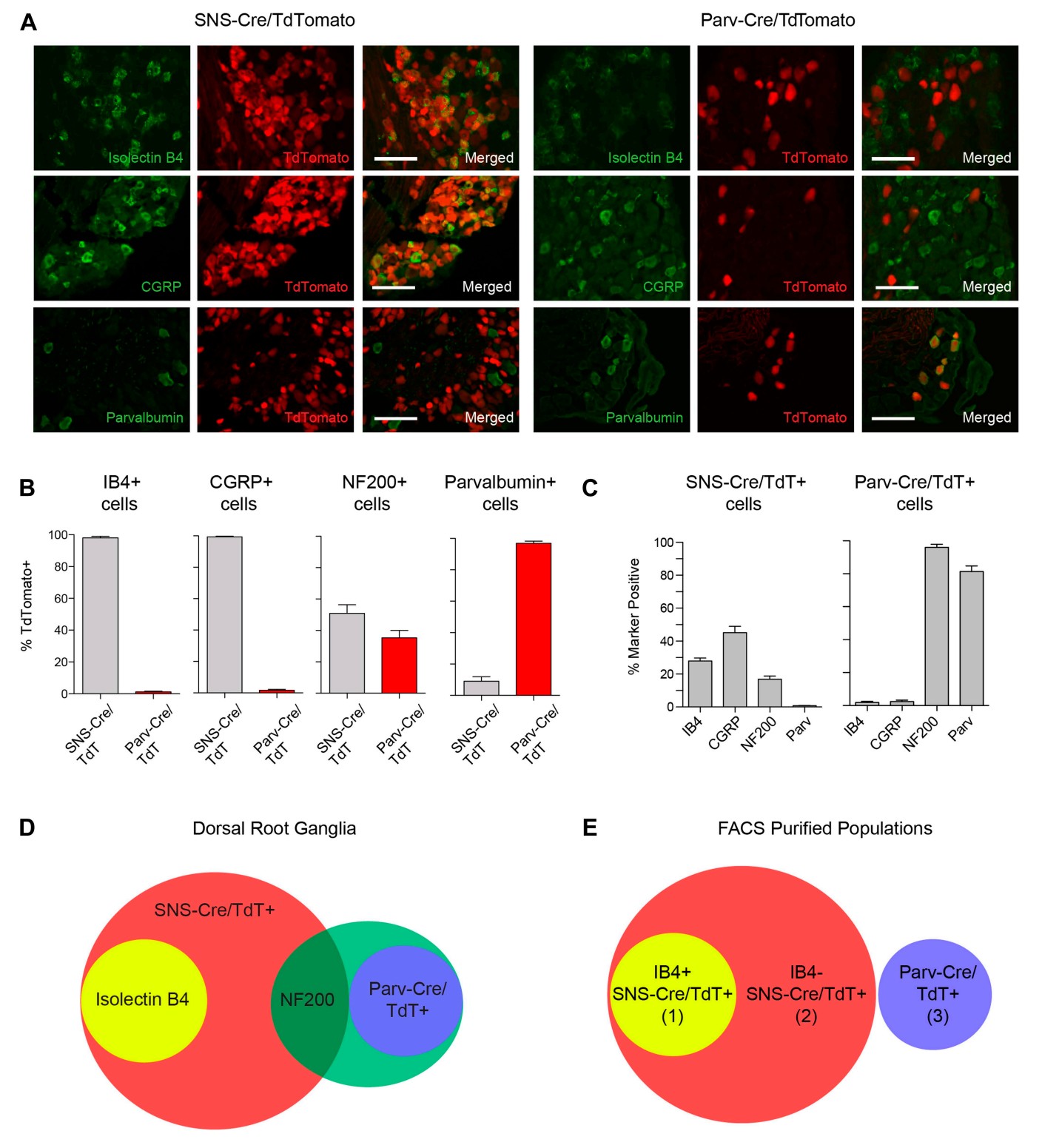

**Figure 1**. Fluorescent characterization of SNS-Cre/TdTomato and Parv-Cre/TdTomato DRG populations. (**A**) SNS-Cre/TdTomato and Parv-Cre/TdTomato lumbar DRG sections imaged for TdTomato (red), IB4-FITC, anti-CGRP, or anti-Parvalbumin (green). Scale bars, 50 μm. (**B–C**) Proportions of IB4+, CGRP+, NF200+, Parvalbumin+ populations expressing SNS-Cre/TdTomato or Parv-Cre/TdTomato, and converse TdTomato proportions expressing each co-stained marker (mean ± s.e.m., n = 8–20 fields from 3 animals). (**D**) Venn diagram depicting distinct DRG populations as labeled by Isolectin B4, NF200, and

*Figure 1. Continued on next page*

*Figure 1. Continued*

TdTomato populations. (**E**) For transcriptional profiling, three non-overlapping DRG populations were FACS purified: IB4+SNS-Cre/TdTomato+, IB4−SNS-Cre/TdTomato+, and Parv-Cre/TdTomato+ cells.
The following figure supplement is available for figure 1:

**Figure supplement 1**. SNS-Cre/TdTomato and Parv-Cre/TdTomato DRG and spinal cord characterization.

labeled 12.5 ± 1.7% DRG neurons, indicating that the majority of primary afferents are included within these two populations. For transcriptome profiling analysis, we purified three non-overlapping sets of DRG neurons: (1) IB4+SNS-Cre/TdTomato+, (2) IB4−SNS-Cre/TdTomato+ and (3) Parv-Cre/TdTomato+ neurons (Venn Diagram, *Figure 1E*).

## Electrophysiology of somatosensory subsets

We analyzed the electrophysiological characteristics of the TdTomato-labeled populations using whole cell patch clamp recordings. Resting membrane potential was similar between SNS-Cre/TdT+ (−56 ± 5.2 mV) and Parv-Cre/TdT+ cells (−60 ± 5 mV). Analyzing firing characteristics, SNS-Cre/TdT+ neurons displayed broad, TTX-resistant action potentials, while Parv-Cre/TdT+ neurons all showed narrow, TTX-sensitive action potentials (*Figure 2A*). These differences were reflected in significant differences in action potential half-width (p = 0.0001 by t-test, *Figure 2B*). Parv-Cre/TdT+ neurons also showed significantly larger capacitance than SNS-Cre/TdT+ neurons (p = 0.0017 by t-test, *Figure 2C*). These differences in firing properties are likely due to distinct ion channel expression patterns. TTX-resistant action potentials are characteristics of Nav1.7 and Nav1.8 nociceptor-lineage neurons (*Bean, 2007*; *Dib-Hajj et al., 2010*; *Shields et al., 2012*). Thus, both anatomically and by neurophysiology, these two lineage reporter mice labeled distinct DRG subsets.

## FACS purification of DRG neuron populations

We performed FACS purification of distinct neuronal populations isolated from both adult (7–20 week old) male and female mice. To avoid multiple rounds of amplification of small quantities of RNA, which would arise from less-abundant neuronal populations such as Parv-cre/TdT+, we chose to pool DRGs from cervical to lumbar regions (C1-L6). DRG cells were enzymatically dissociated and subjected to flow cytometry following DAPI staining to exclude dead cells, and gating on TdTomato<sup>hi</sup> populations (*Figure 3*). This allowed for purification of TdTomato+ neuronal somata with minimal contamination from fluorescent axonal debris and non-neuronal cells (*Figure 3A*). Analysis of our flow cytometry data showed SNS-Cre/TdT+ vs Parv-Cre/TdT+ DRG cells matched the proportions ascertained by NeuN co-staining in DRG sections (*Figure 3B*). It also illustrates that a large percentage of DAPI− live cells are non-neuronal. IB4-FITC surface staining allowed us to simultaneously purify the distinct IB4+ and IB4− subsets within the SNS-Cre/TdT+ population (*Figure 3C*). Forward and side scatter light scattering properties reflect cell size and internal complexity, respectively. SNS-Cre/TdT+ neurons displayed significantly less forward scatter and side scatter than Parv-Cre/TdT+ neurons (*Figure 3—figure supplement 1*). For RNA extraction, DRG populations were sorted directly into Qiazol to preserve transcriptional profiles at the time of isolation.

## Transcriptional profile comparisons of purified neurons vs whole DRG

In total, 14 somatosensory neuron samples were FACS purified consisting of 3–4 biological replicates/neuron population (*Table 1*). We also analyzed RNA from whole DRG tissue for comparison with the purified neuron samples. Because of the small numbers of cells from individual sensory ganglia and to eliminate the need for significant non-linear RNA amplification, total DRGs from three mice were pooled for each sample; following purification, RNA was hybridized to Affymetrix (Santa Clara, CA) microarray genechips for transcriptome analysis.

   Transcriptome comparisons showed few molecular profile differences between biological replicates, but very large inter-population differences (*Figure 3—figure supplement 2*). Importantly, whole DRG molecular profiles differed substantially from the FACS purified neurons. Myelin associated transcripts (Mpz, Mag, Mpz, Pmp2) that are expressed by Schwann cells, for example, showed significantly higher expression in whole DRG tissue than in purified subsets when expressed as

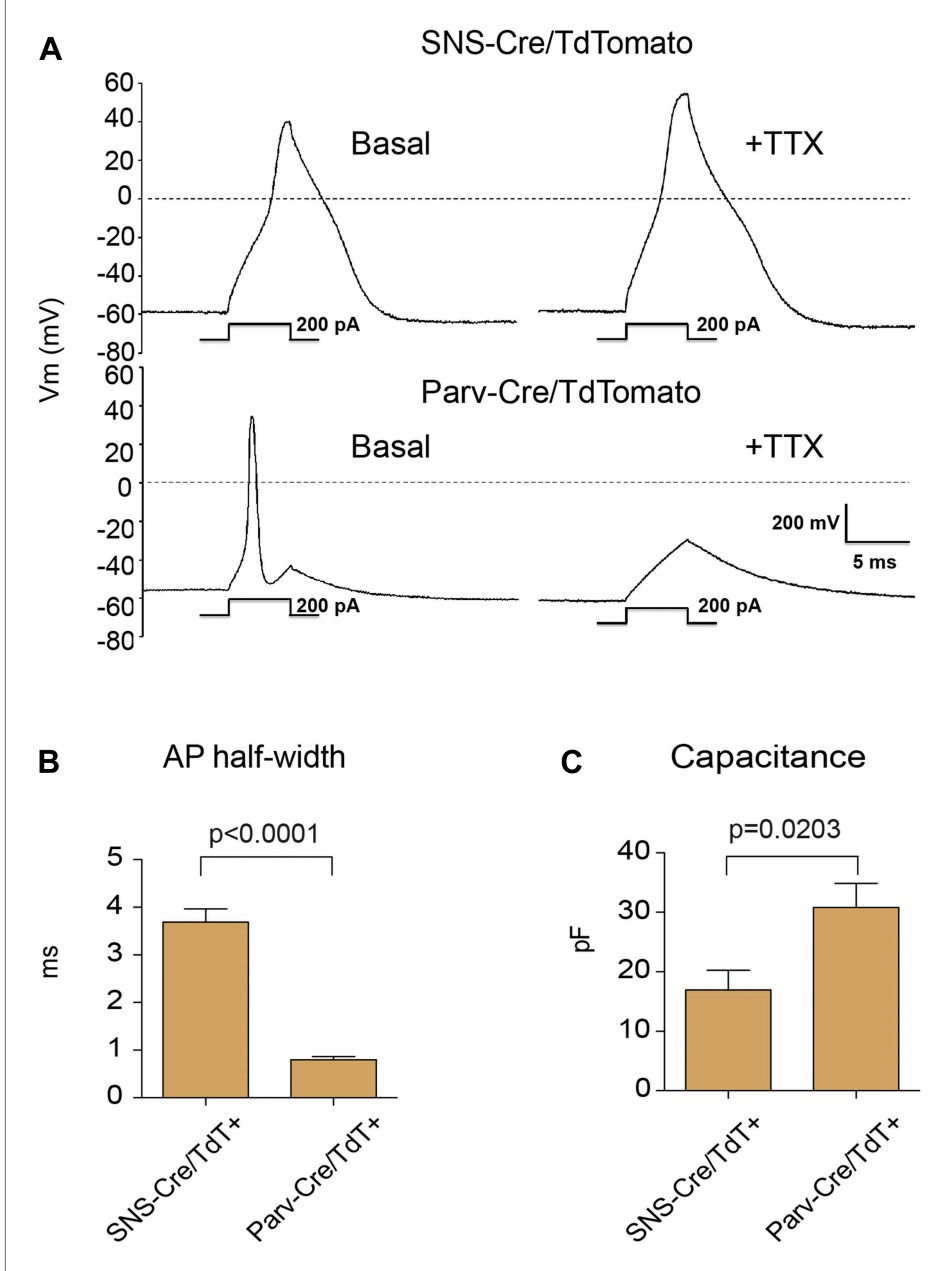

**Figure 2**. Electrophysiological properties of SNS-Cre/TdTomato and Parv-Cre/TdTomato neurons. Whole cell current clamp recordings were conducted on SNS-Cre/TdTomato and Parv-Cre/TdTomato neurons in response to 200 pA injection. (**A**) Representative action potential waveforms before and after application of 500 nM TTX. (**B–C**) Statistical comparisons of action potential (AP) half-widths and capacitances between sensory populations (SNS-Cre/TdT[+], n = 13; Parv-Cre/TdT[+], n = 9; p-values by Student's t test).

absolute robust multi-array average normalized expression levels (***Figure 3—figure supplement 2***). Known nociceptor-associated transcripts (Trpv1, Trpa1, Scn10a, Scn11a) were enriched in SNS-Cre/TdT[+] profiles, and known proprioceptor-associated transcripts (Pvalb, Runx3, Etv1, Ntrk3) were enriched in Parv-Cre/TdT[+] profiles (***Figure 3—figure supplement 2***). Fold-change vs Fold-change plots illustrated the transcriptional differences between purified neurons and whole DRG RNA (***Figure 3—figure supplement 2***), supporting the validity of FACS purification to analyze distinct somatosensory populations compared to whole tissue analysis, which includes mixtures of several neuron populations and many non-neuronal cells.

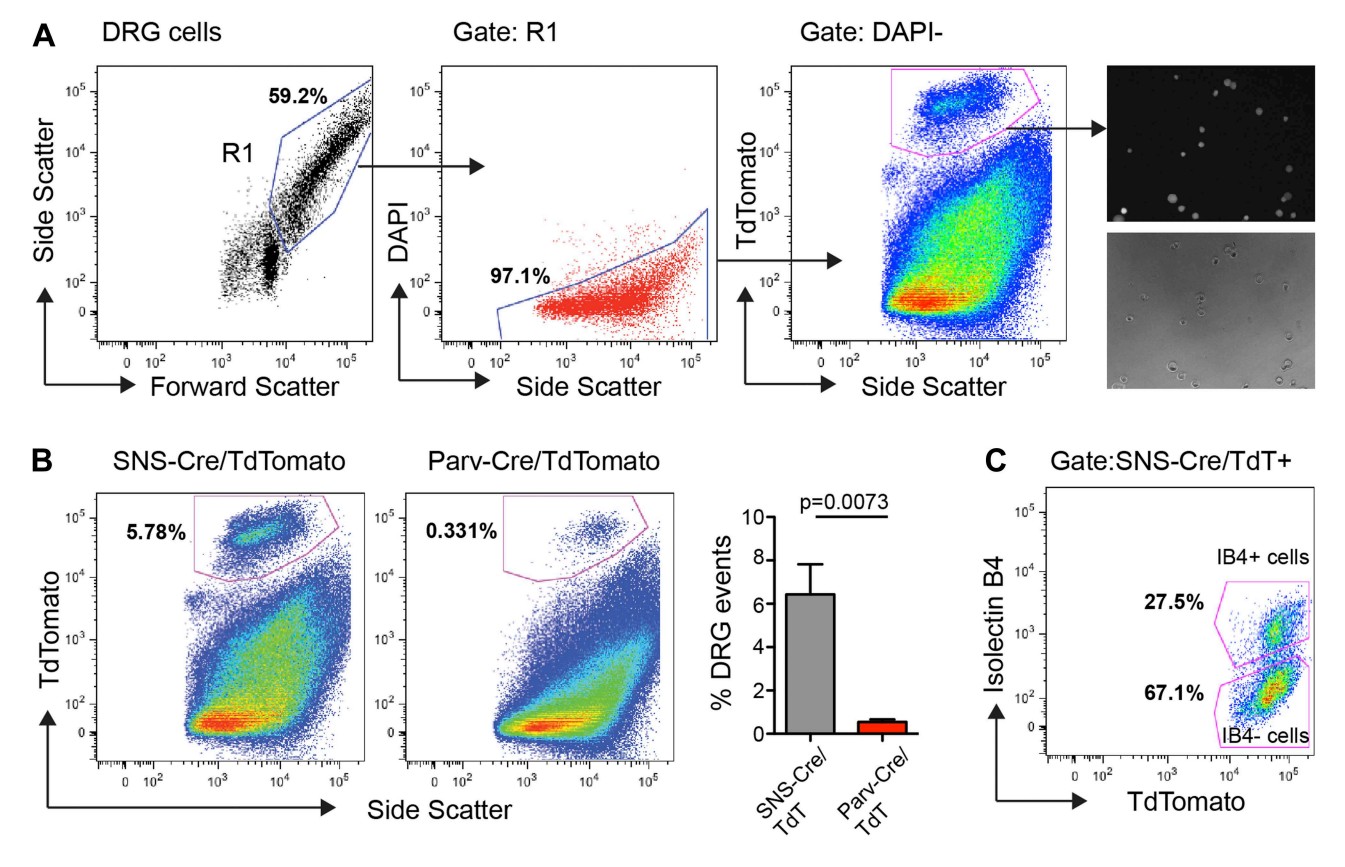

**Figure 3**. FACS purification of distinct somatosensory neuron populations. (**A**) Mouse DRG cells were stained with DAPI and subjected to flow cytometry. After gating on large cells by forward and side scatter (R1), dead cells were excluded by gating on the DAPI⁻ events; Next, TdTomato (hi) events were purified. Following purification, fluorescence and DIC microscopy show that the majority of sorted neurons are TdTomato⁺ (images on right). (**B**) Representative FACS plots of Parv-Cre/TdTomato⁺ and SNS-Cre/TdTomato⁺ DRG populations. Right, quantification of proportions of DAPI⁻ events in the DRG constituting each neuron population (n = 5 SNS-Cre/TdTomato mice, n = 4 Parv-Cre/TdTomato mice; p-values, Student's *t* test; Error bars, mean ± s.e.m.). (**C**) Representative FACS plot shows relative percentages of IB4-FITC surface stained and IB4⁻ neuronal populations among the total SNS-Cre/TdTomato (hi) gate.

The following figure supplements are available for figure 3:

**Figure supplement 1**. Flow cytometric sorting and analysis of TdTomato⁺ neurons.

**Figure supplement 2**. Transcriptome analysis of purified neuronal samples relative to whole DRG tissues.

## Hierarchical clustering and principal components analysis

Hierarchical clustering of molecular profiles from IB4⁺SNS-Cre/TdT⁺, IB4⁻SNS-Cre/TdT⁺, and Parv-Cre/TdT⁺ neuron populations revealed a distinct segregation of these three DRG neuronal subsets, and large blocks of transcripts were enriched for each population (Heat-map, *Figure 4A*). Principal Components Analysis (PCA) showed clustering of samples into distinct groups. IB4⁻SNS-Cre/TdT⁺ neurons differed from Parv-Cre/TdT⁺ neurons along Principal Component 2 (14.49% variation, *Figure 4B*); IB4⁺ and IB4⁻SNS-Cre/TdT⁺ neurons differed along Principal Component 3 (2.58% variation, *Figure 4B*).

## Somatosensory transcript expression across neuronal subsets

We next analyzed gene expression patterns for 36 key known functional mediators of somatosensation (*Figure 5*). The IB4⁺ and IB4⁻ SNS-Cre/TdTomato⁺ neuronal subsets were enriched for the TRP channels, neuropeptides, and G-protein coupled receptors (GPCRs) that are involved in thermosensation,

**Table 1.** Transcriptional samples analyzed in this study

| Sample name | Sample description | Type | n |
|---|---|---|---|
| SNS-Cre/TdT[+] | SNS-Cre/TdTomato[+] FACS purified neurons | Neuron population | 4 |
| Parv-Cre/TdT[+] | Parv-Cre/TdTomato[+] FACS purified neurons | Neuron population | 4 |
| IB4[+]SNS-Cre/TdT[+] | IB4[+]SNS-Cre/TdT[+] FACS purified neurons | Neuron population | 3 |
| IB4[−]SNS-Cre/TdT[+] | IB4[−]SNS-Cre/TdT[+] FACS purified neurons | Neuron population | 3 |
| Whole DRG | Homogenized DRG tissue | Whole tissue | 3 |
| IB4[+]SNS-Cre/TdT[+] (individual neurons) | IB4[+]SNS-Cre/TdT[+] FACS purified single cells | Single cells | 132 |
| IB4[−]SNS-Cre/TdT[+] (individual neurons) | IB4[−]SNS-Cre/TdT[+] FACS purified single cells | Single cells | 110 |
| Parv-Cre/TdT[+] (individual neurons) | Parv-Cre/TdT[+] FACS purified single cells | Single cells | 92 |

In this study, we performed microarray profiling of FACS purified neuron populations, DRG tissue, and single neuron samples. This table summarizes the sample names, descriptions, types, and numbers of samples analyzed. For neuron populations and whole DRG tissue, each biological replicate consisted of pooled total DRG cells from n = 3 animals.

nociception, and pruriception. B-type natriuretic polypeptide b (Nppb), recently identified to mediate itch signaling (*Mishra and Hoon, 2013*), was highly expressed by IB4[−]SNS-Cre/TdT[+] neurons (>800 normalized expression), while gastrin-releasing peptide (GRP), also linked to pruriception (*Sun and Chen, 2007*), was not expressed at detectable levels in any of the purified subsets (<100 normalized expression). Piezo2 (Fam38b), a mechanosensory ion channel (*Coste et al., 2010*; *Maksimovic et al., 2014*; *Woo et al., 2014*), was highly expressed in all somatosensory subsets (>4000 normalized expression), with enrichment in SNS-Cre/TdT[+] relative to Parv-Cre/TdT[+] neurons. By contrast, Trpc1, a channel linked to cutaneous mechanosensation (*Garrison et al., 2012*) was enriched in Parv-Cre/TdT[+] neurons, indicating a potential role in proprioception. C-tactile afferent markers Slc17a8 (Vglut3) and Th (Tyrosine hydroxylase) (*Seal et al., 2009*; *Li et al., 2011*) were enriched in IB4[−]SNS-Cre/TdT[+] neurons, while Mrgprb4 (*Vrontou et al., 2013*) was enriched in IB4[+]SNS-Cre/TdT[+] neurons. Mrgprd and Runx1 were enriched in IB4[+]SNS-Cre/TdT[+] neurons, which are known markers of non-peptidergic nociceptors (*Chen et al., 2006*; *Wang and Zylka, 2009*). Expression of neutrophic factor receptors (Ntrk1, Ntrk2, Ntrk3, Gfra2, Gfra3, Ret) also showed distinct segregation patterns among the IB4[+]SNS-Cre/TdT[+], IB4[−]SNS-Cre/TdT[+] and Parv-Cre/TdT[+] populations. Pvalb, Cadherin 12 (Cdh12), Vglut1 (Slc17a7), and transcription factors (Runx3, Etv1, Etv4) were highly enriched in Parv-Cre/TdT[+] neurons relative to the other two subsets. The distribution of these known mediators or markers of somatosensory function reveals differences and similarities between the three populations that reflect their functional specialization and modality responsiveness.

## Functional neuronal mediators segregate across somatosensory subsets

We next focused our analysis on the expression patterns of those families of genes that mediate different general neuronal functions. Neurons exhibit specific firing properties due to the coordinated activity of different voltage-gated ion channels (*Bean, 2007*; *Dib-Hajj et al., 2010*; *Dubin and Patapoutian, 2010*). We found that many voltage-gated sodium, calcium, potassium, and chloride channels were differentially expressed in the three purified DRG populations (*Figure 6A–D*). Focusing on sodium channels, Scn9a (Nav1.7), Scn10a (Nav1.8), and Scn11a (Nav1.9) were enriched both in the IB4[+] and IB4[−]SNS-Cre/TdT[+] populations (*Figure 6A*), agreeing with known roles in nociception (*Dib-Hajj et al., 2010*). Scn1a (Nav1.1), Scn8a (Nav1.6), and sodium channel beta subunits Scn1b, Scn4b were mainly expressed in Parv-Cre/TdT[+] neurons (*Figure 6A*). Voltage-gated calcium channels, including L-type, N-type, and T-type channels, also showed differential expression (*Figure 6B*). SNS-Cre/TdT[+] neurons were highly enriched for Cacna2d1 (α2δ1) and for Cacna2d2 (α2δ2), the pharmacological targets of gabapentin and pregabalin (*Wang et al., 1999*; *Field et al., 2006*; *Patel et al., 2013*); unexpectedly, Parv-Cre/TdT[+] neurons were enriched for Cacna2d3 (α2δ3) (*Figure 6B*), which contributes to heat nociception via supraspinal expression (*Neely et al., 2010*). Voltage-gated potassium channels showed perhaps the most striking expression patterns across somatosensory subsets (Top 60 most variably expressed shown in *Figure 6C*). Kcns1 (Kv9.1), where a common variant is

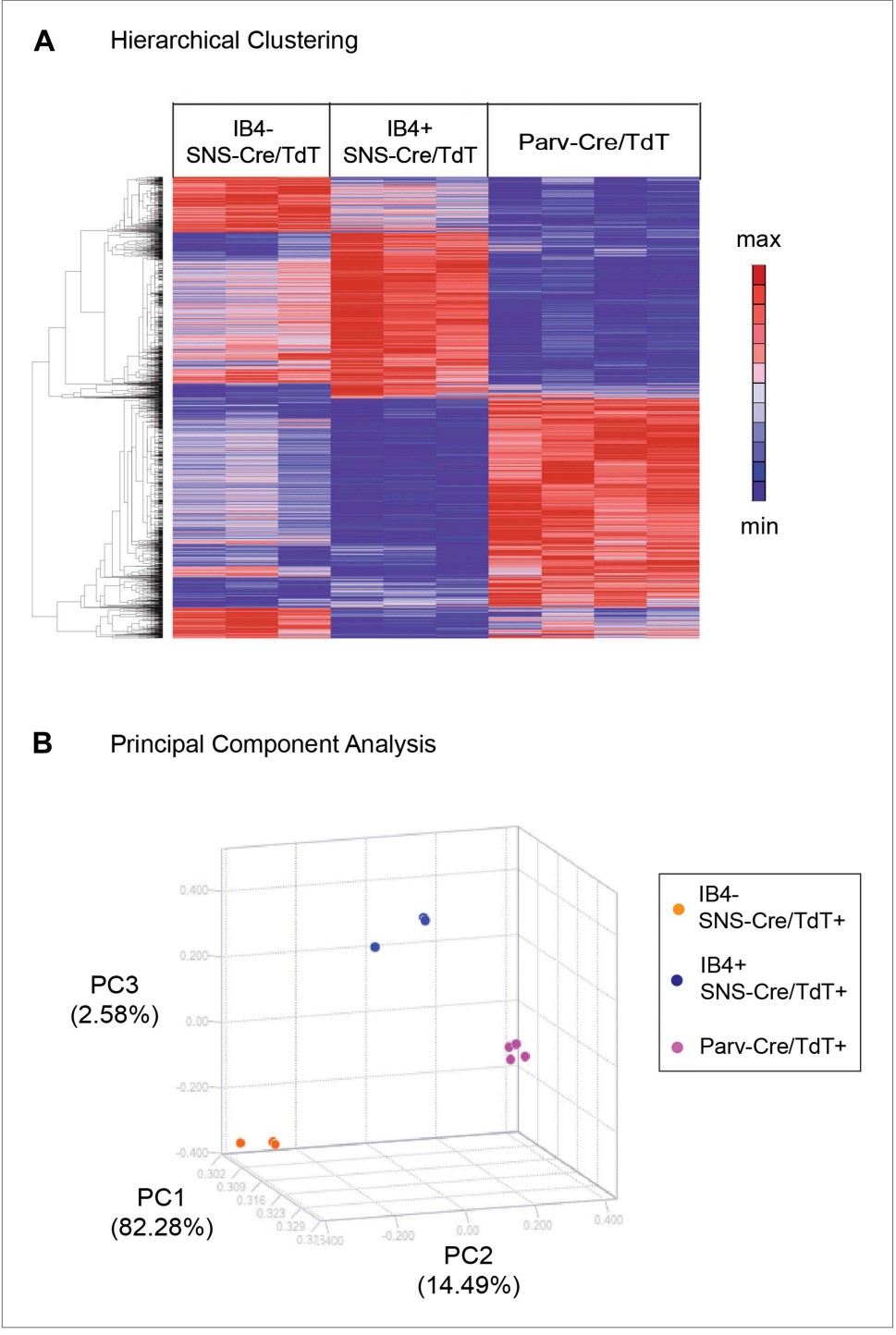

**Figure 4**. Hierarchical clustering and principal components analysis of transcriptomes. (**A**) Hierarchical clustering of sorted neuron molecular profiles (top 15% probesets by coefficient of variation), showing distinct groups of transcripts enriched in IB4+SNS-Cre/TdT+, IB4−SNS-Cre/TdT+, and Parv-Cre/TdT+ neuron populations. (**B**) Principal component analysis shows distinct transcriptome segregation for the purified populations along three principal components axes.

associated with increased pain and whose down-regulation in large sensory neurons is associated with increased neuropathic pain (*Costigan et al., 2010*; *Tsantoulas et al., 2012*), was expressed in Parv-Cre/TdT+ neurons (*Figure 6C*). The IB4+SNS-Cre/TdT+, IB4−SNS-Cre/TdT+, and Parv-Cre/TdT+

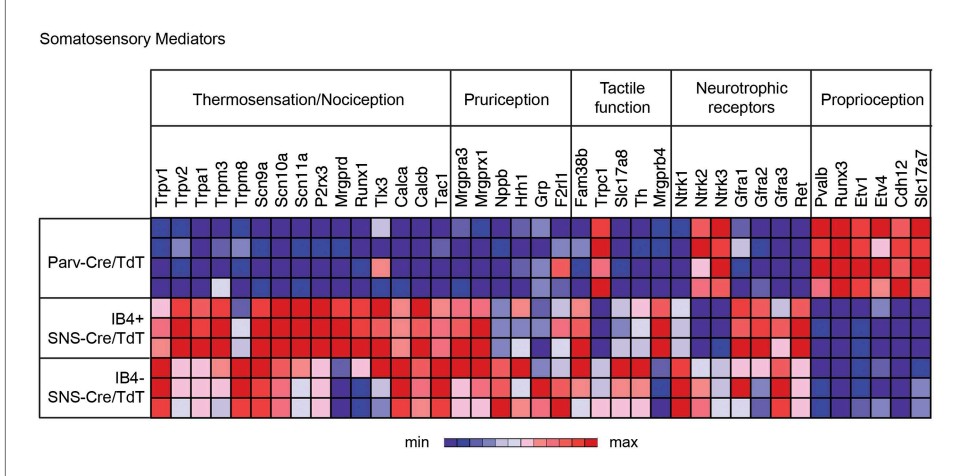

**Figure 5**. Functional somatosensory mediators show clustered gene expression across purified DRG populations. Heat-map showing relative transcript levels for known somatosensory mediators plotted across IB4+SNS-Cre/TdTomato+, IB4−SNS-Cre/TdTomato+, and Parv-Cre/TdTomato+ purified neuron transcriptomes (rows show individual samples; columns are specific transcripts). Genes were grouped based on known roles linked to thermosensation/nociception, pruriception, tactile function, neurotrophic receptors, and proprioception.

populations each showed distinct enrichment patterns for potassium channel genes, most of which have not yet been analyzed yet in terms of somatosensory function. Voltage-gated chloride channels also showed distinct expression patterns, with differential regulation of Clcn and Tweety family ion channel transcripts (**Figure 6D**). Surprisingly, the $Ca^{2+}$ activated chloride channel Ano1 (Anoctamin 1), which has recently been linked to heat nociception (**Cho et al., 2012**), was absent in SNS-Cre/TdT+ populations but present in Parv-Cre/TdT+ neurons (**Figure 6D**).

Transient receptor potential (TRP) channels, ligand-gated ion channels, and G-protein coupled receptors (GPCRs) are integral in the detection of specific environmental stimuli. These different types of molecular transducers showed substantial differential expression across the three purified DRG populations (**Figure 6E** and **Figure 7A–B**). In our dataset, IB4−SNS-Cre/TdT+ neurons were enriched for specific TRP channels (Trpv1, Trpm8, Trpc7, Trpm6), while IB4+SNS-Cre/TdT+ neurons were enriched for others (Trpv2, Trpm4, Trpa1, Trpm3, Trpc6, Trpc5, Trpc3), and only a few TRP channels showed expression in Parv-Cre/TdT+ neurons (Trpm2, Trpc1) (**Figure 6E**). Ligand-gated ion channels also play key roles in nociception or other somatosensory functions. We found diverse expression patterns for HCN channels, P2X channels, 5-HT receptors (Htr3a, Htr3b) ionotropic glutamate receptors, GABA receptors, and Glycine receptors across the neuronal populations (Top 60 most variably expressed ligand-gated channels, **Figure 7A**). GPCRs, including Mas-related GPCRs, muscarinic glutamate receptors, neuropeptide receptors, as well as some orphan receptors showed significant expression in different somatosensory subsets (Top 60 most variably expressed GPCRs, **Figure 7B**). Taken together, these data show complex patterns of ligand-gated molecular transducer expression that could play roles in functional specialization and signaling.

We also found that many transcription factors were differentially expressed across these three neuron populations (Top 60 most variably expressed TFs, **Figure 7C**). Many of these have not yet been explored in the somatosensory system, and could play roles in neuronal differentiation and maintenance of cell-type specification during adulthood. For example, Klf7 and Isl2 were expressed at high levels and enriched in SNS-Cre/TdT+ neurons (>1.5-fold, p < 0.01, >5000 expression). Based on this analysis, these two transcription factors have now been used to facilitate the trans-differentiation of fibroblasts into nociceptor-like neurons in conjunction with other transcription factors (**Wainger et al., 2014**).

## Pair-wise enrichment analysis of neuron populations

To obtain statistically significant and unbiased enrichment analysis, we next performed pair-wise comparisons of the three major neuronal subclasses. We first compared SNS-Cre/TdTomato and

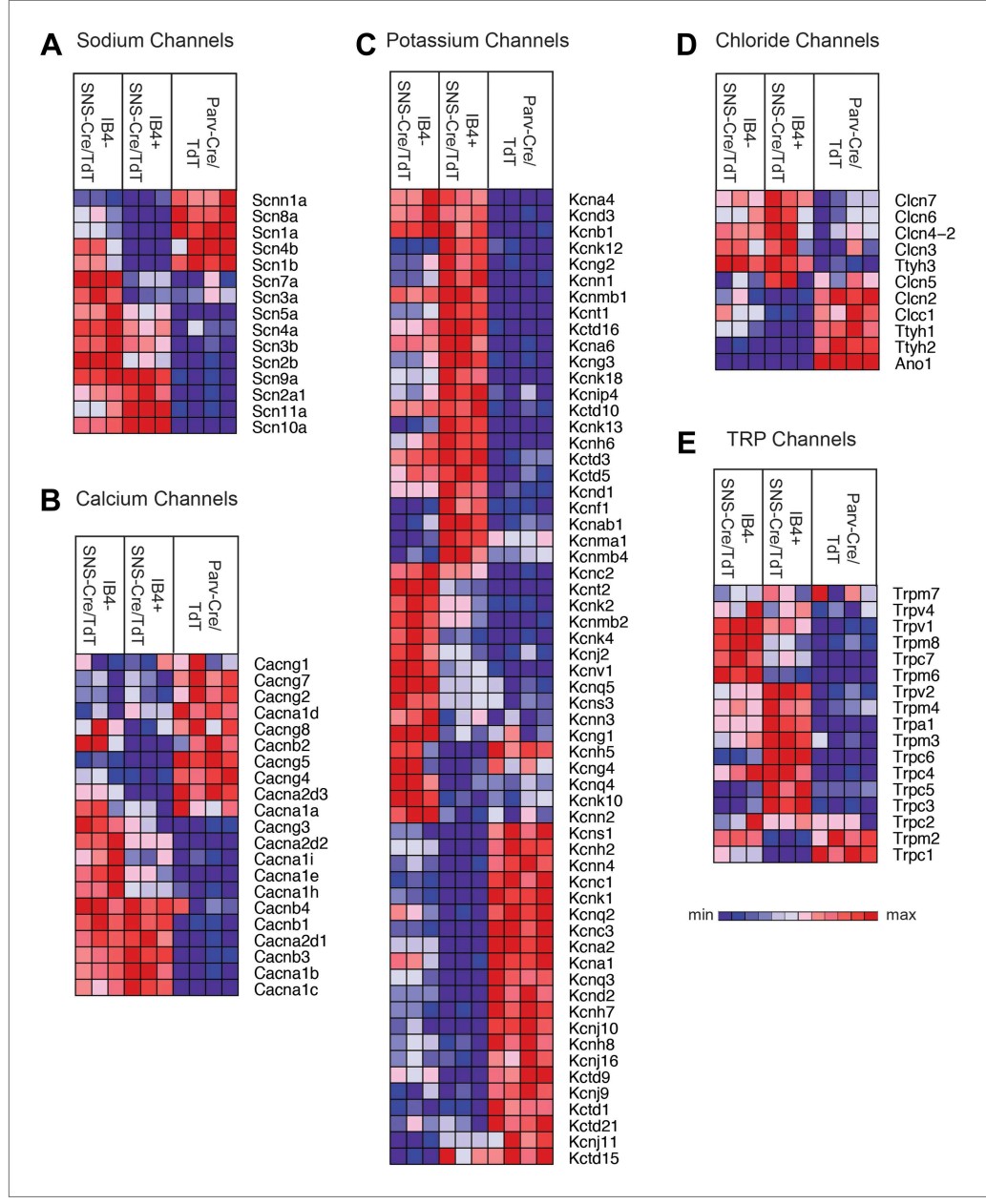

**Figure 6**. Heat-map distribution of voltage-gated and TRP channels across neuronal subsets. Expression patterns of different sub-types of voltage-gated ion channels and transient receptor potential (TRP) channels were hierarchically clustered and analyzed across IB4+SNS-Cre/TdT+, IB4−SNS-Cre/TdT+ and Parv-Cre/TdT+ purified neuron samples (columns are individual samples, heat-maps). (**A**) Sodium channel levels, (**B**) calcium channel levels, (**C**) potassium channel levels (top 60 differentially expressed transcripts by CoV), (**D**) chloride channel levels, and (**E**) TRP channel levels are plotted as heat-maps. For **A**–**E**, plotted transcripts show minimum expression >100 in at least one neuronal subgroup.

Parv-Cre/TdTomato neurons, yielding many differentially expressed (DE) genes in each neuron subset (SNS-Cre/TdT+: 907 genes, Parv-Cre/TdT+: 774 genes, p < 0.05, twofold; ***Figure 8A***, ***Supplementary file 1***). Specific Gene Ontology (GO) categories and Kyoto Encyclopedia of Genes and Genomes (KEGG) pathways were significantly enriched (***Figure 8B***). The most differentially expressed GO categories were GO:0006816~calcium ion transport and GO:0006813~potassium ion transport. Thus, we focused on calcium ion channels and potassium ion channels using volcano plot comparisons (***Figure 8C***). SNS-Cre/TdT+ vs Parv-Cre/TdT+ neurons showed differential regulation of various calcium channels

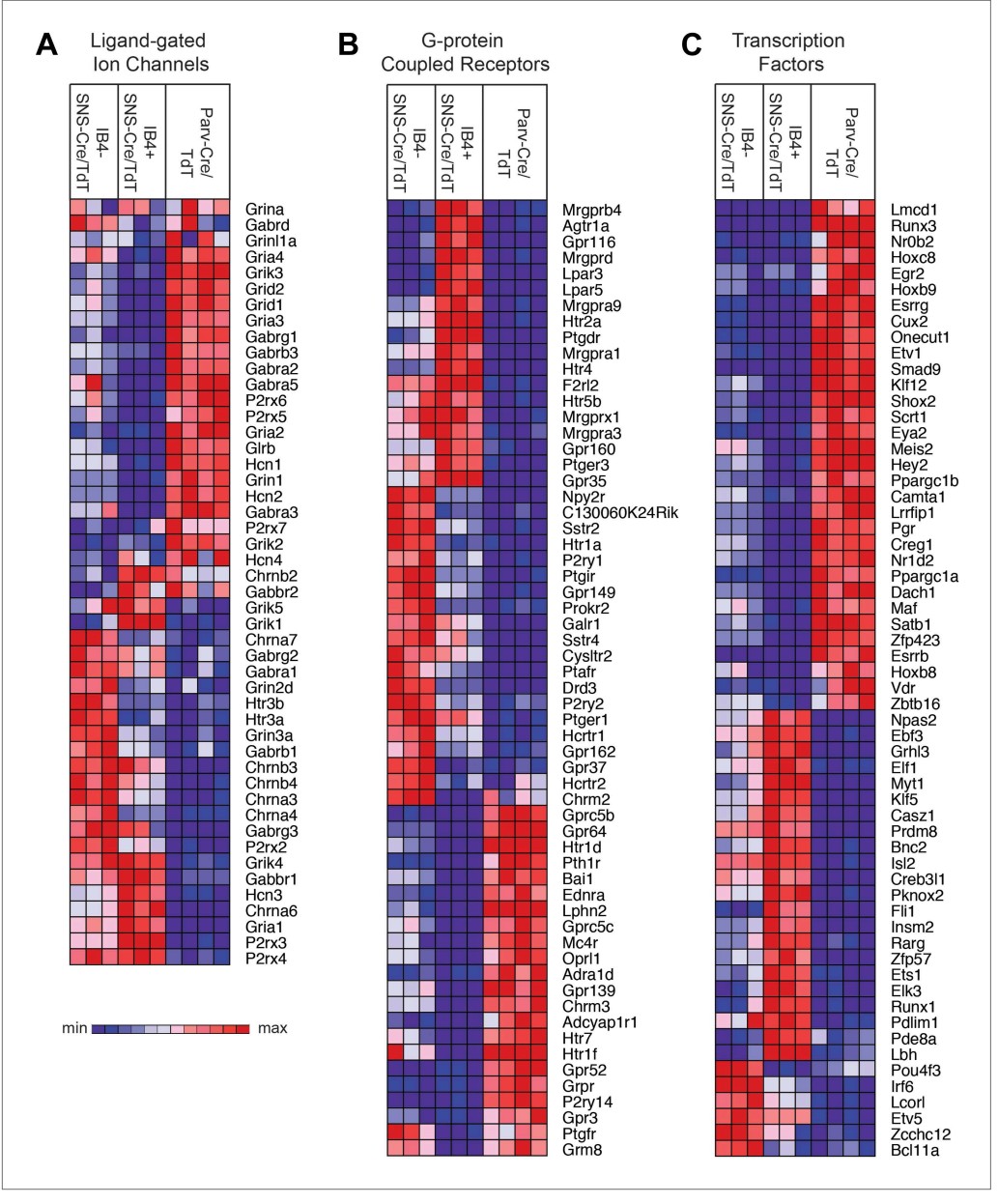

**Figure 7**. Heat-map distribution of ligand-gated ion channels, G-protein coupled receptors, and transcription factors across neuronal subsets. (**A**) Expression patterns of ligand-gated ion channels, including glutamatergic, chlorinergic, HCN, P2X channels, were analyzed by hierarchical clustering (columns are individual samples). (**B**) Differentially expressed G-protein coupled receptors (GPCRs) were clustered and plotted across sensory subsets (Top 60 by CoV are shown). (**C**) Differentially expressed transcription factors were clustered and plotted across sensory subsets as a heat-map (Top 60 by CoV are shown). For **A–C**, plotted transcripts show minimum expression >100 in at least one neuronal subgroup.

(SNS-Cre/TdT$^+$: 9 genes, Parv-Cre/TdT$^+$: 3 genes, twofold, $p < 0.01$, *Figure 8C-i*) and potassium channels (SNS-Cre/TdT$^+$: 15 genes, Parv-Cre/TdT$^+$: 12 genes, twofold, $p < 0.01$, *Figure 8C–I*). Based on statistical criteria of fold-change >2, $p < 0.01$, all differentially expressed TRP channels were enriched only in SNS-Cre/TdT$^+$ neurons, which may relate to their importance in thermosensation and nociception (8 genes, *Figure 8C-iii*).

In a second pairwise comparison, IB4$^+$ were compared with IB4$^-$ SNS-Cre lineage neurons (*Figure 9*). This analysis yielded 258 significantly enriched transcripts in IB4$^+$ vs 492 in IB4$^-$ neurons

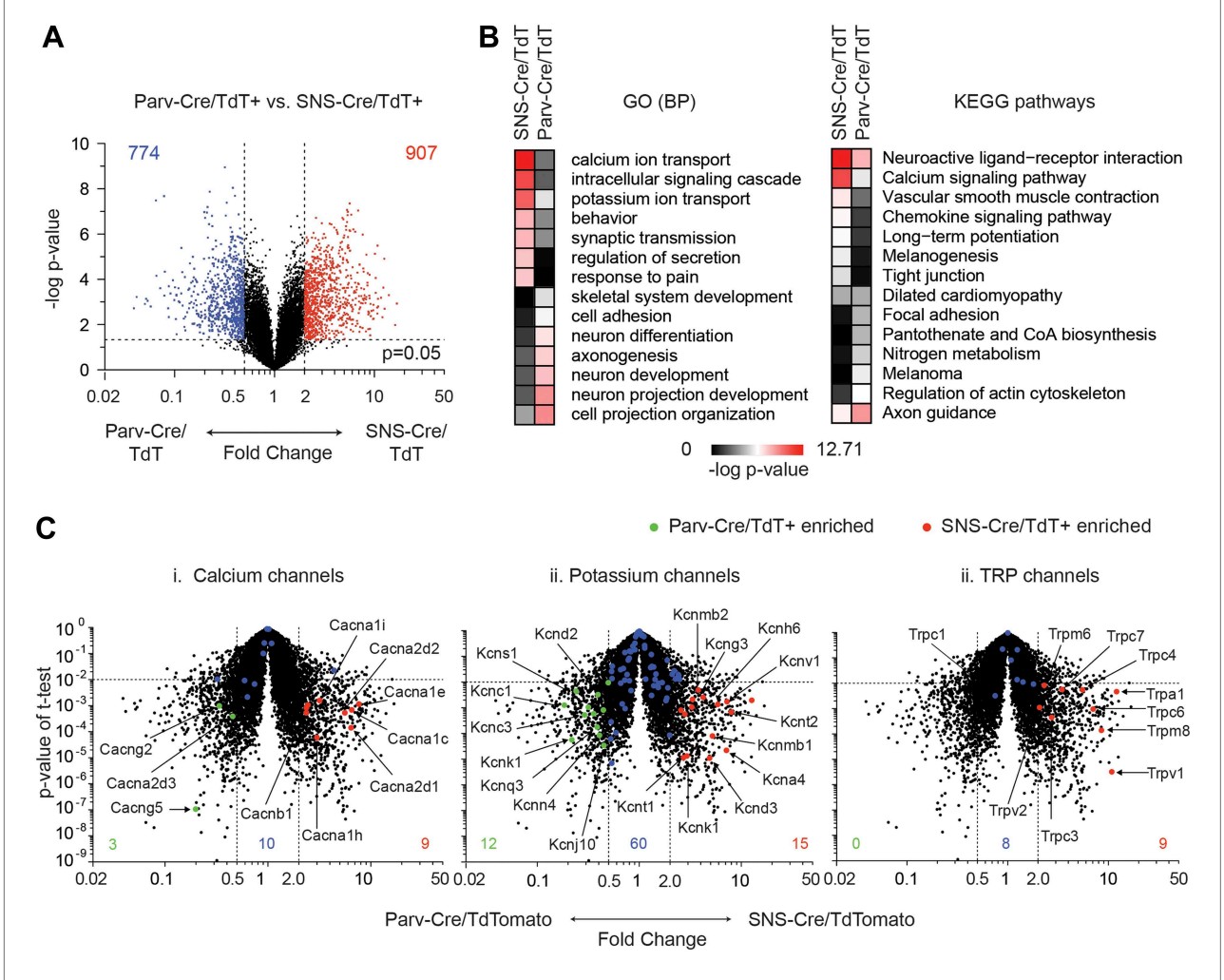

**Figure 8**. Differential volcano plot analysis of SNS-Cre/TdTomato vs Parv-Cre/TdTomato transcriptomes. (**A**) Pairwise comparison of SNS-Cre/TdT⁺ vs Parv-Cre/TdT⁺ profiles showing differentially expressed (DE) transcripts as a volcano plot (blue transcripts, Parv-Cre/TdT enriched; red, SNS-Cre/TdT enriched, twofold, p < 0.05). (**B**) Most enriched Gene ontology (GO) categories and Kyoto Encyclopedia of Genes and Genomes (KEGG) pathways in SNS-Cre/TdT vs Parv-Cre/TdT enriched transcripts, plotted as heat-map of −log (p-value). (**C**) Volcano plots depicting (**i**) calcium channels, (**ii**) potassium channels, and (**iii**) TRP channels expression differences between populations. Individual transcripts highlighted (red, SNS-Cre/TdT⁺ enriched; green, Parv-Cre/TdT⁺ enriched; blue, not significantly different: twofold, p < 0.01).

(twofold, p < 0.05, *Figure 9A*, *Supplementary file 2*). GO categories differentially regulated between IB4⁺ and IB4⁻ subsets included those for ion transport, cell adhesion, and synaptic transmission (*Figure 9B*). Volcano plot analysis shows significant differential expression of ion channels between these two subsets (IB4⁻SNS-Cre/TdT⁺: 29 genes, IB4⁺SNS-Cre/TdT⁺: 16 genes, p < 0.05; twofold, *Figure 9C-i*). P2rx3 (P2X3) and Scn11a (Nav1.9), ion channels known to mark non-peptidergic nociceptors, were enriched 1.8-fold and twofold, respectively in IB4⁺SNS-Cre/TdT⁺ neurons. Interestingly, we found even greater enrichment for Trpc3 (7.98-fold) and Trpc6 (7.67-fold) in this subset. Focusing on cell adhesion, volcano plots showed differential enrichment for Nrxn3, Nrcam, and Ncam2 in IB4⁻SNS-Cre/TdT⁺ neurons, and Cdh1, Pvrl1 in IB4⁺SNS-Cre/TdT⁺ neurons (*Figure 9C-ii*). Next, we focused on GPCR expression differences (*Figure 9C-iii*). Mrgprd, a widely used marker of non-peptidergic neurons (*Wang and Zylka, 2009*), was enriched 20.6-fold in IB4⁺ neurons. Interestingly, we found several GPCRs that were enriched as Mrgprd in IB4⁺SNS-Cre/TdT⁺ neurons but have not yet been characterized for function in this subset, including Agtr1a (20.4-fold), Gpr116 (15.7-fold), Lpar3 (11.8-fold), and Lpar5 (12.6-fold).

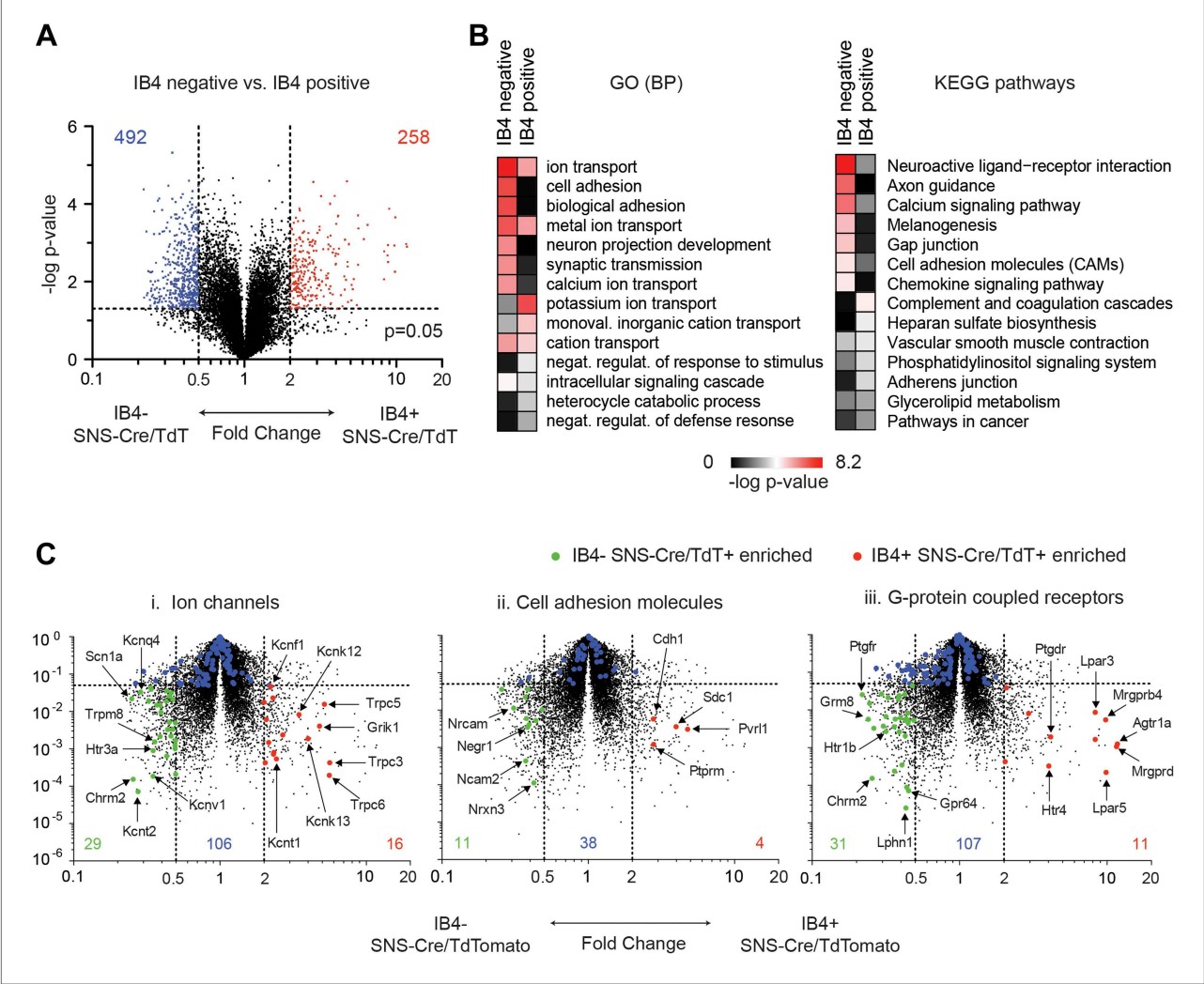

**Figure 9**. Differential volcano plot analysis of IB4$^+$ and IB4$^-$ SNS-Cre/TdTomato subset transcriptomes. (**A**) Pairwise comparison of IB4$^+$SNS-Cre/TdT$^+$ vs IB4$^-$SNS-Cre/TdT$^+$ neuronal profiles show differentially expressed (DE) genes by volcano plot (blue, IB4$^+$ enriched; red, IB4$^-$enriched, twofold, p < 0.05). (**B**) Top Gene ontology (GO) categories of biological processes (BP) and Kyoto Encyclopedia of Genes and Genomes (KEGG) pathways for IB4$^+$SNS-Cre/TdT$^+$ and IB4$^-$SNS-Cre/TdT$^+$ enriched transcripts, plotted as heat-maps of −log (p-value). (**C**) Volcano plots showing differential expression of (**i**) ion channels, (**ii**) cell adhesion molecules, and (**iii**) G-protein coupled receptors between neuronal populations (red, IB4$^+$ enriched transcripts; green, IB4$^-$ enriched; blue, not significantly different: twofold, p < 0.01).

## Single cell analysis reveals log-scale gene expression heterogeneity

We next performed single cell level transcriptional analysis of the three globally characterized DRG populations using Fluidigm parallel qRT-PCR gene expression technology. Distinct transcriptional hallmarks for each FACS purified population were first defined by their differential expression in the microarray datasets (threefold enrichment, *Figure 10*). Taqman assays were chosen corresponding to these enriched markers, and including two housekeeping genes (Gapdh and Actb), a complete group of 80 assays was used for single cell expression profiling (*Table 2*). We first used these assays to analyze 100-cell and 10-cell FACS sorted groups of each neuronal population (*Figure 10—figure supplement 1*), confirming the enrichment of various marker transcripts.

We next FACS sorted individual IB4$^+$SNS-Cre/TdT$^+$, IB4$^-$SNS-Cre/TdT$^+$, and Parv-Cre/TdT$^+$ neurons into 96-well plates for Fluidigm analysis. A total of 334 individual neurons were purified and analyzed (IB4$^+$SNS-Cre/TdT$^+$ cells, n = 132; IB4$^-$SNS-Cre/TdT$^+$ cells, n = 110; and Parv-Cre/TdT$^+$ cells, n = 92, *Table 1*).

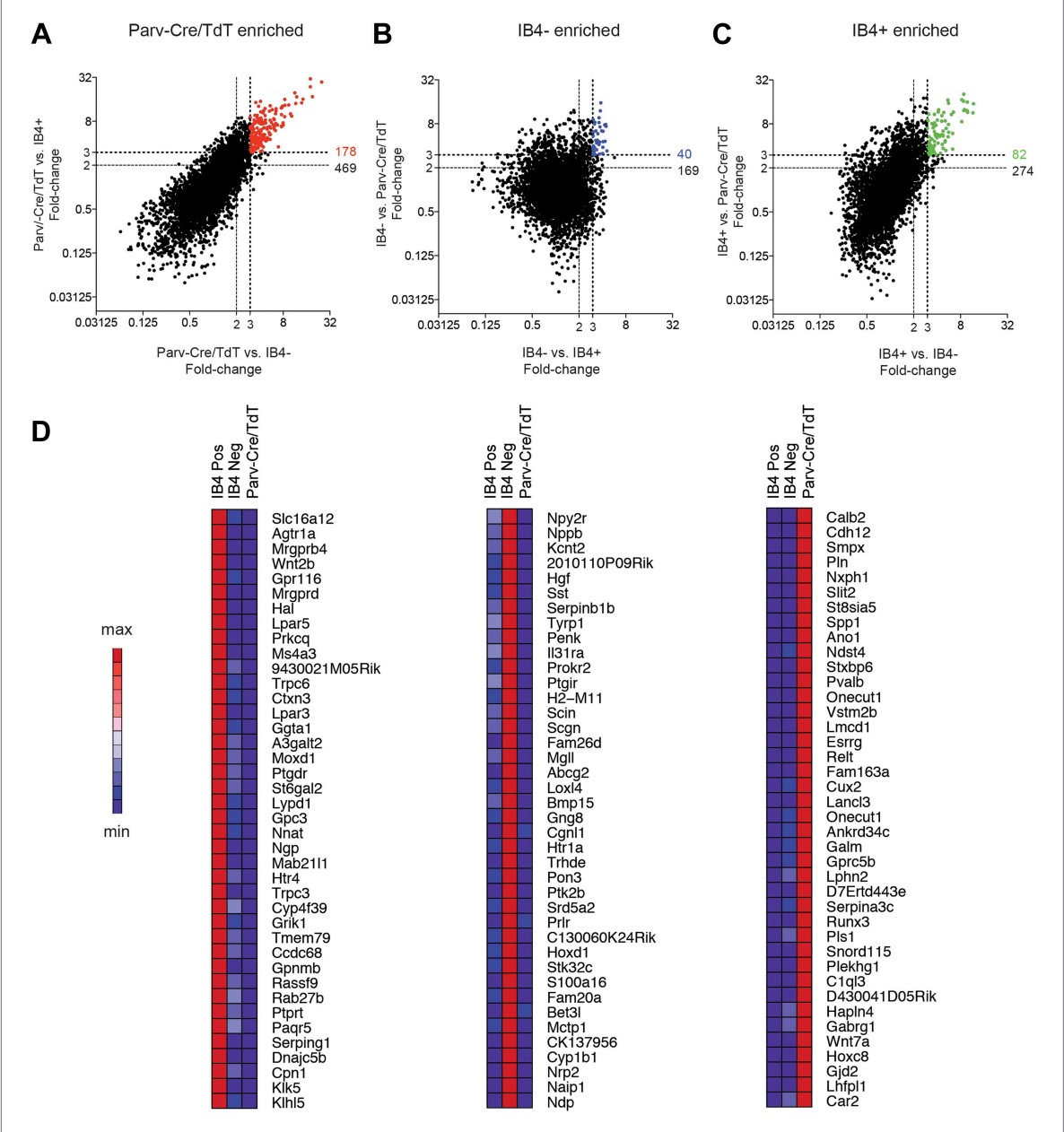

**Figure 10**. Analysis of most enriched marker expression by IB4+, IB4− SNS-Cre/TdTomato and Parv-Cre/TdTomato+ populations. (**A–C**) Fold-change/ fold-change comparisons illustrate most differentially enriched genes in each subset (highlighted in color are threefold and twofold enriched numbers). (**D**) Heat-maps showing relative expression of the top 40 transcripts enriched in each of the three neuronal subsets (>threefold), ranked by product of fold-change differences.

The following figure supplement is available for figure 10:

**Figure supplement 1**. Fluidigm analysis of 100 and 10 cell-samples.

We found that the expression levels for specific transcripts across single cell datasets often displayed a log-scale continuum (*Figure 11*). Some transcripts were highly enriched in one subset of single cells (e.g., Mrgprd, Trpv1, P2rx3), but were often nonetheless expressed at detectable levels in other neuronal groups. This continuum of gene expression made it difficult to set 'thresholds' for assigning the presence or absence of a particular transcript. Thus, we focused our definition of distinct

**Table 2.** Taqman assays used for single cell transcriptional profiling

| SNS-Cre/TdT+ enriched (vs Parv-Cre/TdT) | IB4+ SNS-Cre/TdT+ enriched | IB4− SNS-Cre/TdT+ enriched | Parv-Cre/TdT+ enriched |
|---|---|---|---|
| Trpv1 | Mrgprd | Smr2 | Pvalb |
| Trpa1 | P2rx3 | Npy2r | Runx3 |
| Scn10a | Agtr1a | Nppb | Calb2 |
| Scn11a | Prkcq | Kcnv1 | Slit2 |
| Isl2 | Wnt2b | Prokr2 | Spp1 |
| Kcnc2 | Slc16a12 | Ptgir | Ano1 |
| Galr1 | Lpar3 | Th | Stxbp6 |
| Car8 | Lpar5 | Il31ra | St8sia5 |
| Chrna3 | Trpc3 | Ntrk1 | Ndst4 |
| Atp2b4 | Trpc6 | Bves | Esrrb |
| Aqp1 | Moxd1 | Kcnq4 | Esrrg |
| Chrna6 | A3galt2 | Htr3a | Gprc5b |
| Pde11a | St6gal2 | S100a16 | Car2 |
| MrgprC11 | Mrgprb4 | Pou4f3 | Pth1r |
| Syt5 | Mrgprb5 | Cgnl1 | Wnt7b |
| Gfra3 | Ptgdr | | Kcnc1 |
| Klf7 | Ggta1 | | Etv1 |
| Cysltr2 | Grik1 | | Pln |
| Irf6 | Mmp25 | | Cdh12 |
| Prdm8 | Casz1 | | |
| Etv5 | Bnc2 | | |
| Stac | Klf5 | | |
| | Lypd1 | | |
| Housekeeping genes | | | |
| Gapdh | Actb | | |

To perform Fluidigm single cell analysis, Taqman assays were chosen to cover four categories of population-enriched transcripts first identified by microarray whole transcriptome analysis: (1) SNS-Cre/TdT+ (total population) enriched markers (vs Parv-Cre/TdT+ neurons), (2) IB4+SNS-Cre/TdT+ enriched markers (vs other 2 groups), (3) IB4−SNS-Cre/TdT+ markers (vs other 2 groups), and (4) Parv-Cre/TdT+ markers (vs other 2 groups). Taqman assays for housekeeping genes Gapdh and Actb were also included.

subgroups not by absolute proportion of positive gene expression but by correlative and aggregate analysis. Other transcripts (e.g., Nppb, Runx3, Cdh12) showed expression patterns restricted in one population and were not present in other populations.

## Hierarchical clustering of single cell data reveals distinct subgroups

Spearman-rank hierarchical clustering was performed on the Fluidigm expression data normalized to gapdh expression (columns represent single cells, *Figure 12*). This analysis revealed a high degree of heterogeneity of transcriptional expression across the three DRG populations. The vast majority of single cells showed distinct patterns of expression of at least one neuronal transcript, including voltage-gated ion channels (Scn10a, Scn11a, Kcnc2, Kcnv1), ligand-gated channels (P2rx3, Trpv1, Trpa1), and Parvalbumin (Pvalb) indicating minimal amplification noise (*Figure 12—figure supplement 1*). Unbiased spearman rank analysis revealed seven distinct neuronal subgroups (*Figure 12*). Six out of seven groups had 24 or more individual cells (group I, 115 cells; group II, 50 cells; group III, 4 cells; group IV, 24 cells; group V, 24 cells; group VI, 24 cells; group VII, 93 cells). We chose one level of sample segregation to analyze, but other cellular subclasses are likely present at lower levels of clustering (*Figure 12*). Importantly, when hierarchical clustering was performed on data normalized to

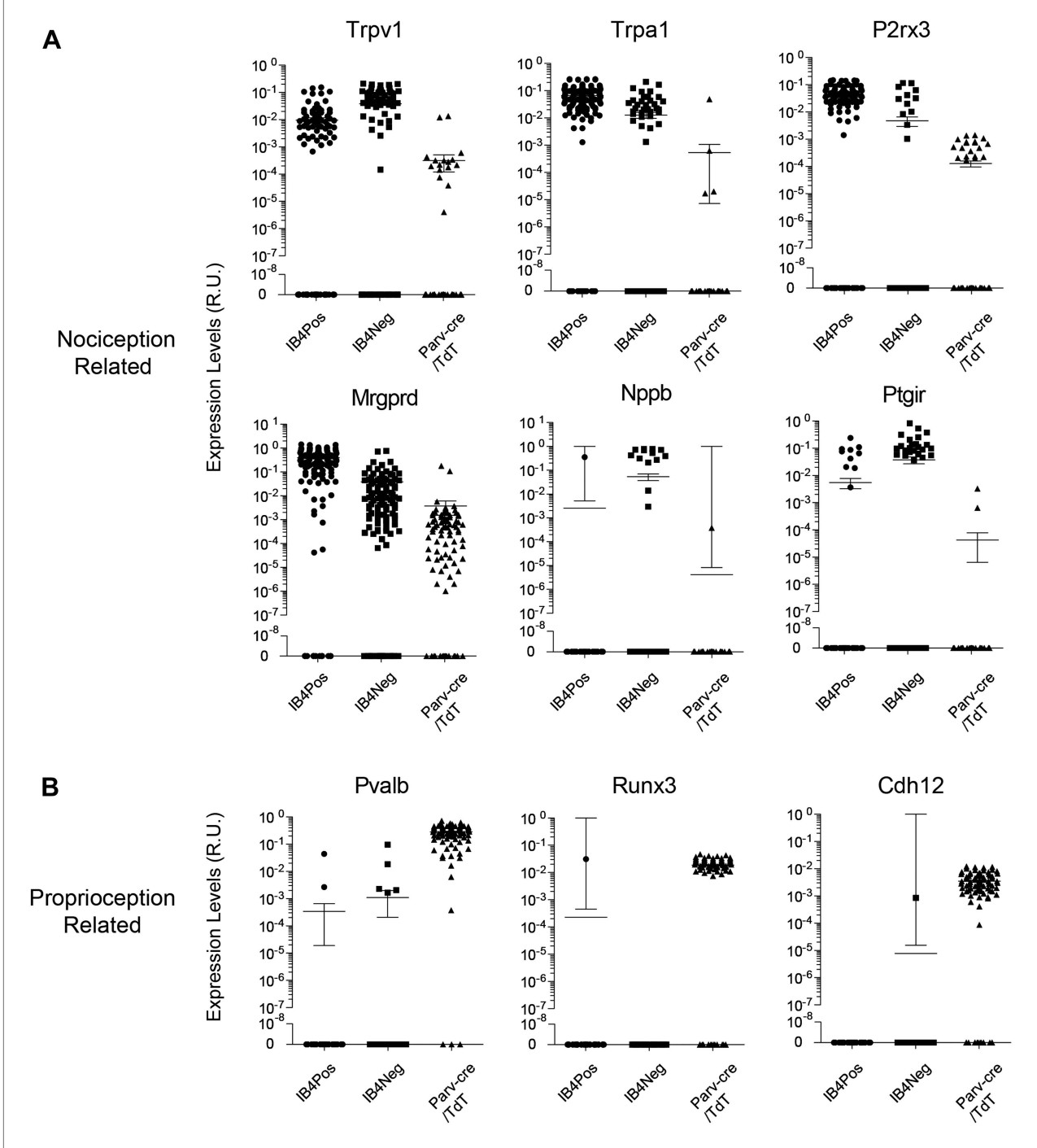

**Figure 11**. Single cell transcript levels show log-scale distribution across neuronal populations. Normalized transcript levels in single cells determined by parallel qRT-PCR are plotted on a log-scale comparing IB4+SNS-Cre/TdT+, IB4–SNS-Cre/TdT+, and Parv-Cre/TdT+ cells. (**A**) Nociceptor related transcript levels (Trpv1, Trpa1, Mrgprd, P2rx3, Nppb, Ptgir), (**B**) Proprioception related transcript levels (Pvalb, Runx3, Cdh12). Individual neurons are shown as dots in plots.

Actb, neuronal subgroups based on gapdh normalization segregated in a similar manner (data not shown). Principal components analysis showed distinct separation of the single cell subgroups along different principal components (**Figure 13A**), with Groups I and VII on disparate arms of PC2 (~5% variation), while Group V neurons segregated along PC3 (~1.88% variation). Parv-Cre/TdT+

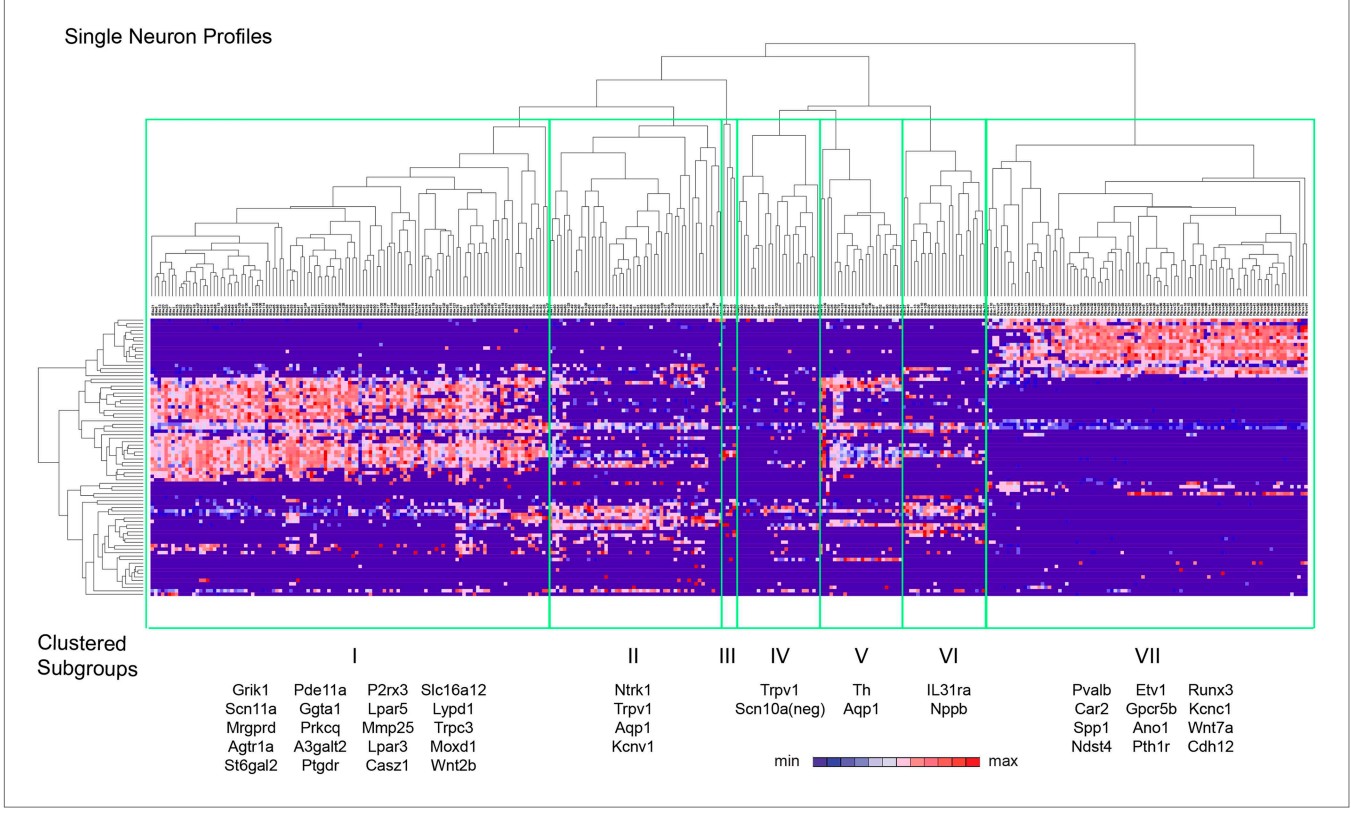

**Figure 12**. Hierarchical clustering analysis of single cell qRT-PCR data reveals distinct neuronal subgroups. Heat-map of 334 single neurons and 80 genes after spearman-rank hierarchical analysis of RT-PCR data (relative gene expression normalized to gapdh). Each column represents a single sorted cell, and each transcript is shown per row. Clustering analysis finds seven distinct subgroups (I, II, III, IV, V, VI, VII). Characteristic transcript expression patterns that delineate each somatosensory subset are written below.

The following figure supplements are available for figure 12:

**Figure supplement 1**. Expression of neuronal-associated transcripts across purified single cell samples by qRT-PCR.

**Figure supplement 2**. Transcript expression levels for characteristic marker genes in single cell neuron Group I and Group VII.

neurons mainly fell within group VII (96.7% of the cells, **Figure 13B**). IB4+SNS-Cre/TdT+ and IB4−SNS-Cre/TdT+ neurons were distributed among subgroups II–VI (**Figure 13B**). Therefore, this analysis has uncovered potentially novel subgroups distributed across the SNS-Cre/TdT+ population that are not captured by the presence or absence of IB4 staining.

## Major characteristics of distinct single cell subgroups

We next analyzed the major characteristics of each DRG single cell subgroup (**Figure 12**). Group I neurons were mostly IB4+ nociceptors enriched for Pr2x3, Scn11a, and Mrgprd, markers for non-peptidergic nociceptors. Our analysis found a large number transcriptional hallmarks for Group I neurons that were as well enriched as the known marker genes, including Grik1, Agtr1a, Pde11a, Ggta1, Prkcq, A3galt2, Ptgdr, Lpar5, Mmp25, Lpar3, Casz1, Slc16a12, Lpyd1, Trpc3, Moxd1, Wnt2b (**Figure 12**, and **Figure 12—figure supplement 2**). Nearest neighbor analysis across all single cells found 13 transcripts with Pearson correlation >0.5 for Mrgprd, further showing a large cohort of genes that segregate in expression within group I neurons (**Figure 14**).

Group II neurons expressed high levels of Ntrk1 (Trka), Scn10a (Nav1.8), and Trpv1. We also found that they expressed significant levels of Aqp1 (Aquaporin 1), and a major proportion of Group II neurons also expressed Kcnv1 (Kv8.1). Group III consisted of only four cells and we thus did not consider it a true neuronal subclass.

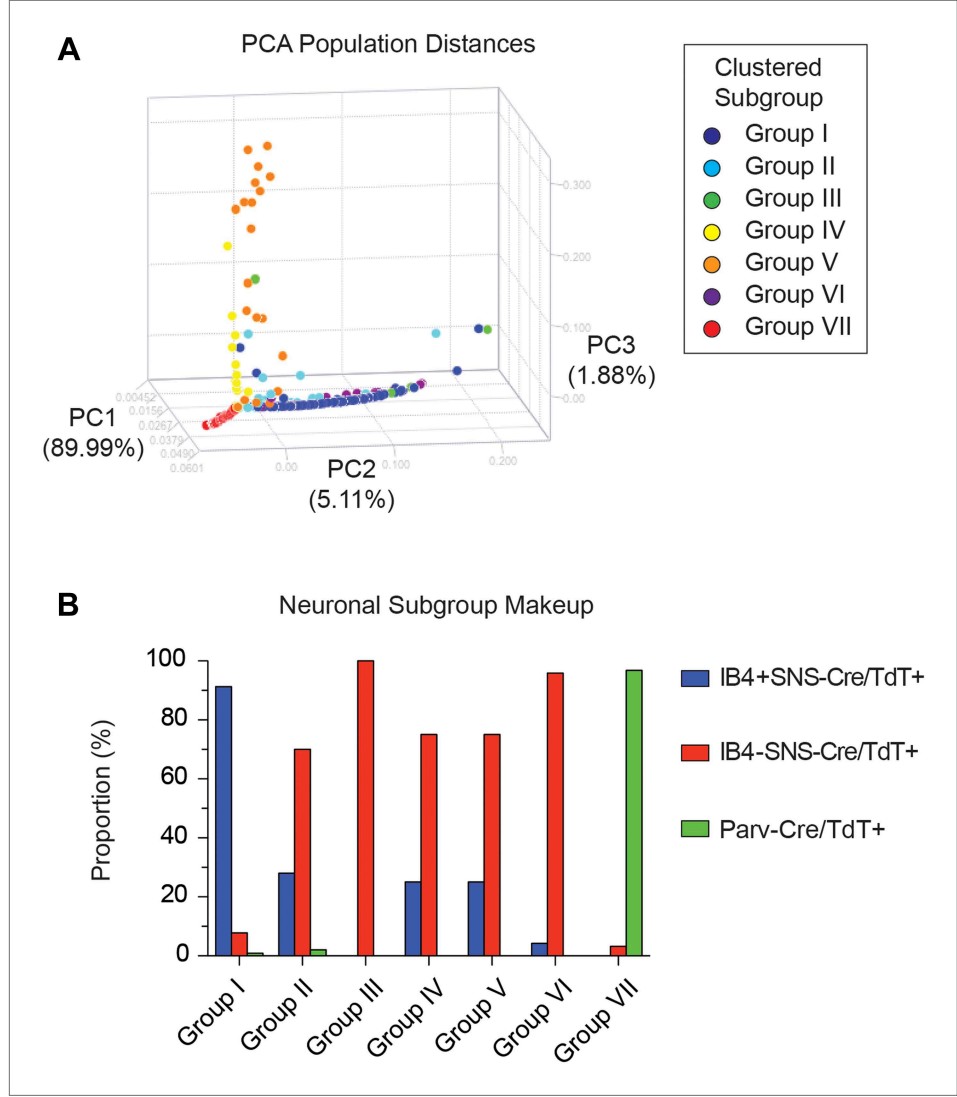

**Figure 13**. Single cell subgroups distribute differentially across originally purified populations. (**A**) Principal Components Analysis of single cell transcriptional data shows distinct segregation of Groups I, V, and VII neurons. (**B**) Proportions of each neuronal subgroup relative to original labeled IB4[+]SNS-Cre/TdTomato[+], IB4[−]SNS-Cre/TdTomato[+], and Parv-Cre/TdTomato[+] neurons.

Group IV neurons were characterized by the absence of Scn10a (Nav1.8) but the presence of Trpv1 expression (*Figure 14—figure supplement 1*). Although Group IV neurons were all labeled by SNS-Cre/TdTomato, they did not all show Scn10a gene expression, likely reflecting transient transcription of this transcript that is shutdown in some neurons during development (*Liu et al., 2010*).

Group V neurons were distinguished by Th (tyrosine hydroxylase) gene expression, a known marker for low-threshold C-mechanoreceptors (*Li et al., 2011*). Triple immunofluorescence with IB4 showed that TH fell mostly within the IB4[−]SNS-Cre/TdT[+] subset (91.4 ± 2.4% TH[+] were IB4[−]SNS-Cre/TdT[+], *Figure 15—figure supplement 1*). Th[+] neurons also expressed high levels of Scn10a (Nav1.8) and Aqp1 (Aquaporin 1), but low/undetectable levels of Ntrk (Trka) and Trpv1 (*Figure 14—figure supplement 1, 2*).

Group VI neurons were a distinct population characterized by co-expression of Nppb and IL31ra (*Figure 14*). Nppb is a neuropeptide mediator of itch signaling from DRG neurons to spinal cord pruritic circuitry (*Mishra and Hoon, 2013*). IL31 is a T cell cytokine associated with pruritus, and DRG neurons express the IL31 receptor (*Bando et al., 2006*; *Sonkoly et al., 2006*) Co-expression of IL31ra

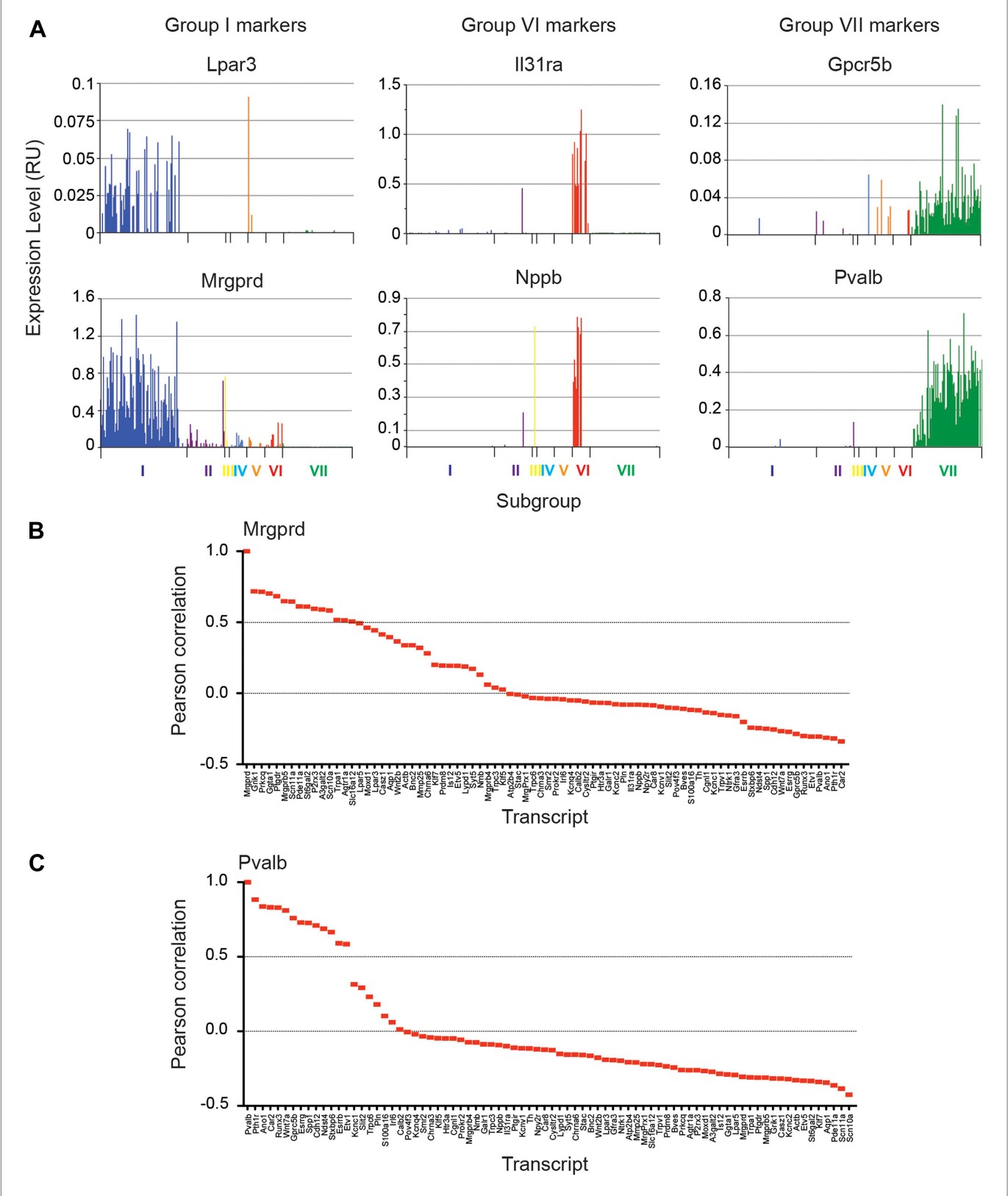

**Figure 14**. Focused analysis of single cell heterogeneity and transcript enrichment in neuronal subgroups. (**A**) Relative expression levels of subgroup specific transcripts in single cells across each neuronal subgroup (each bar = 1 cell). Group I (Lpar3, Mrgprd), group VI (Il31ra, Nppb), and group VII markers (Gpcr5b) show subset enrichment and highly heterogeneous expression at the single cell level. (**B–C**) Nearest neighbor

*Figure 14. Continued on next page*

Figure 14. Continued

analysis by pearson correlation of Mrgprd and Pvalb transcript levels to all 80 probes across the single cell expression dataset was generated. Correlation levels go from left to right.

The following figure supplements are available for figure 14:

**Figure supplement 1**. Defining the transcriptional characteristics of Group I, II, and IV neurons.

**Figure supplement 2**. Expression plots of nociceptor-associated transcripts across single cell transcriptional data.

with Nppb suggests that these neurons may be specialized in mediating itch. Group VI neurons also showed high level expression of Scn10a, Scn11a, and Trpv1 relative to the other subsets (*Figure 14—figure supplement 1, 2*).

Group VII neurons consisted of 93 cells comprising the majority of Parv-Cre/TdT+ sorted single cells (*Figures 12, 13*). These neurons showed high expression of Parvalbumin (Pvalb) and osteopontin (Spp1), Cadherin 12 (Cdh12) as well as proprioceptor associated transcription factors (Etv1, Runx3). Nearest neighbor analysis of Pvalb gene expression showed 13 transcripts with Pearson correlation >0.5 (*Figure 14*). These include a set of distinct ion channels (Kcnc1, Ano1), GPCRs (Pth1r, Gpcr5b), and other genes (Ndst4, Car2, Wnt7a) that have not been functionally characterized in this subset.

### Fluorescence *ISH* analysis of subgroup-specific characteristics

RNA in situ *hybridization* (ISH) was used to confirm specific localization of novel Group I, VI, and VII enriched transcripts (*Table 3*). The Group I marker Lysophosphatidic acid receptor 3 (Lpar3) labeled a subset of SNS-Cre/TdT+ neurons that did not overlap with Parv-Cre/TdT+ expression (*Figure 15*). We found similar results for Prkcq (PKCθ), another Group I marker (*Figure 15—figure supplement 2*). The Group VI marker Il31ra also labeled a distinct subset of SNS-Cre/TdT+ neurons and did not colocalize with Parv-Cre/TdT+ neurons (*Figure 15*). By contrast, the group VII marker Gpcr5b did not stain SNS-Cre/TdT+ neurons but co-localized well with Parv-Cre/TdT+ proprioceptors (*Figure 15*). Double ISH found that itch-related Group VI marker IL31ra did not colocalize with group I markers Prkcq or Lpar3, nor with group VII marker Gpcr5b (*Figure 15*). In confirmation of the Fluidigm data, double ISH found that IL31ra colocalized well with Nppb (*Figure 15*), thus confirming co-expression of two itch-related markers in the same neuronal subset. Thus, expression profiling at single cell resolution reveals an unsuspected degree of complexity of sensory neurons with elucidation of many new markers and of different neuronal subtypes.

### Discussion

Mapping neuronal circuitry and defining the molecular characteristics of specific neurons is critical to understanding the functional organization of the nervous system. The somatosensory system, all of whose primary sensory neurons are of neural crest origin, is highly complex, innervating diverse peripheral tissues and encoding thermal, mechanical, and chemical modalities across a broad range of sensitivities, from innocuous to noxious with different dynamic ranges (*Marmigere and Ernfors, 2007*; *Basbaum et al., 2009*; *Dubin and Patapoutian, 2010*; *Li et al., 2011*). Sensory neurons are currently classified based on myelination and conduction properties (i.e., C-, Aα/β- or Aδ-fibers) or their selective expression of ion channels (e.g., Trpv1, P2rx3, Nav1.8), neurotrophin receptors (e.g., TrkA, TrkB, TrkC, Ret), cytoskeletal proteins (e.g., NF200, Peripherin), and GPCRs (e.g., Mrgprd, Mrgpra3). However, combining these different classification criteria can result in complex degrees of overlaps, making a cohesive categorization of distinct somatosensory populations challenging. Transcriptome-based analysis has become recently a powerful tool to understand the organization of complex populations, including subpopulations of CNS and PNS neurons (*Lobo et al., 2006*; *Sugino et al., 2006*; *Molyneaux et al., 2009*; *Okaty et al., 2009, 2011*; *Lee et al., 2012*; *Mizeracka et al., 2013*; *Zhang et al., 2014*). In this study, we performed cell-type specific transcriptional analysis to better understand the molecular organization of the mouse somatosensory system.

Our population level analysis revealed the molecular signatures of three major classes of somatosensory neurons. There were vast transcriptional differences between SNS-Cre/TdTomato+ and Parv-Cre/TdTomato+ neurons, potentially reflecting their developmental specification into neurons

**Table 3.** RNA in situ *hybridization* probes

| Gene | Forward primer | Reverse primer | Probe length (bp) |
|---|---|---|---|
| Gpcr5b | 5'-ATGTTCCTGGT | 5'-TCACCAATGGTG | 1233 |
| Lpar3 | 5'-TTGTGATCGTCCTGTGCGTG | 5'-GCCTCTCGGTATTGCTGTCC | 870 |
| TdTomato | 5'-ATCAAAGAGTTCATGCGCTTC | 5'-GTTCCACGATGGTGTAGTCCTC | 615 |
| Prkcq | 5'-TCTTGCTGGGTCAGAAGTACAA | 5'-TCTGTGGTTGAGTGGAATTGAC | 919 |
| Nppb | 5'-TGAAGGTGCTGTCCCAGATGATTC | 5'-GTTGTGGCAAGTTTGTGCTCCAAG | 545 |
| Il31ra | 5'-CTCCCCTGTGTTGTCCTGAT | 5'-TTCATGCCATAGCAGCACTC | 559 |

Probesets used for RNA in situ *hybridization* analysis. Listed are gene symbols, sequences for forward and reverse primers, and resulting probe lengths.

with quite different functional attributes and targets. As SNS-Cre is expressed mainly within TrkA-lineage neurons (**Abdel Samad et al., 2010**; **Liu et al., 2010**), while Parv-Cre is expressed mainly in proprioceptor-lineage neurons (**Hippenmeyer et al., 2005**), these two populations reflect archetypical C- and Aα/β-fibers, respectively. Bourane et al previously performed SAGE analysis of TrkA deficient compared to wild-type DRGs, which revealed 240 differentially expressed genes and enriching for nociceptor hallmarks (**Bourane et al., 2007**). Our FACS sorting and comparative population analysis identified 1681 differentially expressed transcripts (twofold), many of which may reflect the early developmental divergence and vast functional differences between these lineages. While C-fibers mediate thermosensation, pruriception and nociception from skin and deeper tissues, Parv-Cre lineage neurons mediate proprioception, innervating muscle spindles and joints (**Marmigere and Ernfors, 2007**; **Dubin and Patapoutian, 2010**). Almost exclusive TRP channel expression in SNS-Cre/TdT$^+$ neurons vs Parv-Cre/TdT$^+$ neurons may relate to their specific thermosensory and chemosensory roles. We also found significant molecular differences between the IB4$^+$ and IB4$^-$ subsets of SNS-Cre/TdT$^+$ neuronal populations. Our analysis identified many molecular hallmarks for the IB4$^+$subset (e.g., Agtr1a, Casz1, Slc16a12, Moxd1) that are as enriched as the currently used markers (P2rx3, Mrgprd), but whose expression and functional roles in these neurons have not yet been characterized. This analysis of somatosensory subsets covered the majority of DRG neurons (~95%), with the exception of TrkB$^+$ Aδ cutaneous low-threshold fibers (**Li et al., 2011**), which are NF200$^+$ but we find are negative for SNS-Cre/TdTomato and Parv-Cre/TdTomato (Data not shown).

Single cell analysis by parallel quantitative PCR of hundreds of neurons demonstrated large heterogeneity of gene expression within the SNS-Cre/TdT$^+$ neuron population, much greater than the current binary differentiation of peptidergic or non-peptidergic IB4$^+$ subclasses. Interestingly, we found a log-scale continuum for many transcripts, including nociceptive genes (e.g., Trpv1, Trpa1) showing high expression in IB4$^+$ and IB4$^-$ subsets and with lower but not absent levels in Parv-Cre/TdT$^+$ cells. This may reflect transcriptional shut-down of genes during differentiation. Unbiased hierarchical clustering analysis of single cell data revealed at least six distinct neuronal subgroups. These findings reveal new molecular characteristics for known neuron populations and also uncover novel neuron subsets: Group I neurons consist of Mrgprd$^+$Nav1.8$^+$P2rx3$^+$Nav1.9$^+$ cells, which are polymodal non-peptidergic C-fibers, for which we identify a panoply of new molecular markers. Group II consists of Trka$^{hi}$Nav1.8$^+$Trpv1$^+$Aquaporin$^+$ neurons, matching known characteristics of thermosensitive C-fibers; many of these expressed Kcnv1. Group V consists of Th$^+$Nav1.8$^+$Trka$^-$Trpv1$^-$ cells, matching characteristics of C-fiber low-threshold mechanoreceptors (C-LTMRs) (**Li et al., 2011**). Group VII consists of Pvalb$^+$Runx3$^+$Etv1$^+$ neurons, which are mostly proprioceptor-lineage neurons for which we identified 12 molecular markers. Lee et al recently performed transcriptome analysis of purified TrkC-lineage proprioceptive neurons in the presence or absence of NT-3 signaling (**Lee et al., 2012**) and we note that Group VII neurons were similar to TrkC lineage cells in gene expression (Pth1r, Runx3, Pvalb). Group IV consists of Trpv1$^+$Nav1.8$^-$ neurons, which may represent a unique functional subgroup; Wood et al found that mice depleted for Nav1.8-lineage neurons retained a TRPV1 responsive subset (**Abrahamsen et al., 2008**). We uncover a new subset of neurons, Group VI, which appears to represent pruriceptive neurons based on their co-expression of IL31ra and Nppb.

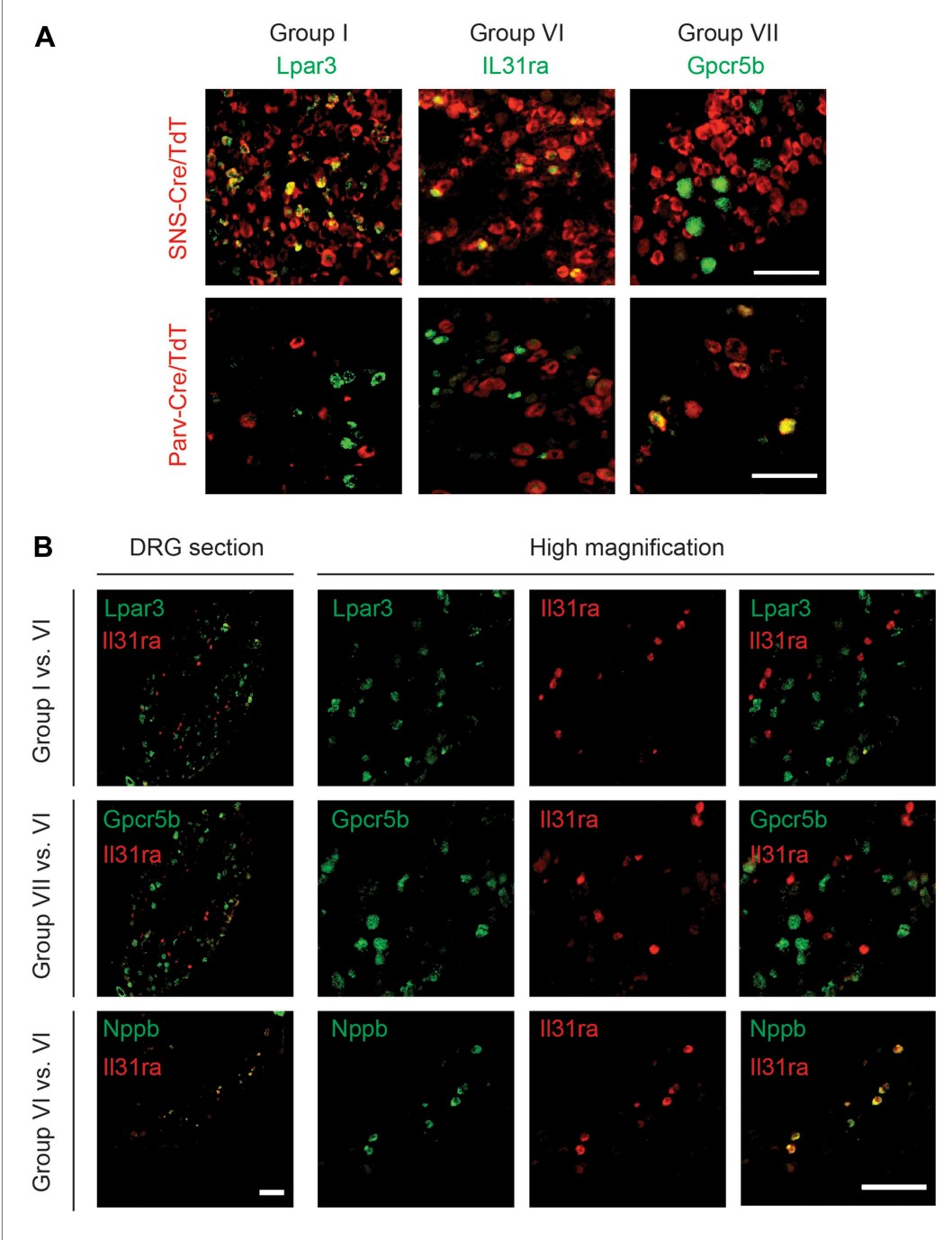

**Figure 15**. DRG subgroups I, VI, and VII characteristics defined by double RNA in situ *hybridization*. (**A**) Double RNA in situ *hybridization* in SNS-Cre/TdTomato and Parv-Cre/TdTomato lumbar DRG sections for TdTomato (red) with Lpar3, Il31ra, or Gpcr5b (green), which are Group I, VI, and VII markers respectively. Lpar3 and IL31ra expression colocalize with SNS-Cre/TdTomato but not Parv-TdTomato, while Gpcr5b colocalizes with Parv-Cre/TdTomato but not SNS-Cre/TdTomato. (**B**) Double in situ *hybridization* in lumbar DRG sections for group VI marker IL31ra vs Group I marker Lpar3, Group VI marker Gpcr5b, or Group VI marker Nppb. Il31ra and Nppb in shown in a distinct subset of DRG neurons. Scale bars, 100 μm.

The following figure supplements are available for figure 15:

**Figure supplement 1**. Immunofluorescence characteristics of DRG subgroup V.

*Figure 15. Continued*

**Figure supplement 2**. Group I marker Prkcq is in a distinct subset of DRG neurons.

While preparing this manuscript, several papers performing expression profiling of postnatal adult somatosensory neurons were published (*Goswami et al., 2014*; *Thakur et al., 2014*; *Usoskin et al., 2014*). We note that each study utilized distinct methodologies from our work: Goswami et al profiled Trpv1-Cre/TdTomato⁺ neurons compared to Trpv1-diptheria toxin depleted whole DRG tissue (*Goswami et al., 2014*). Thakur et al performed magnetic bead selection to remove DRG non-neuronal cells, performing RNA-seq on residual cells enriched for neurons (*Thakur et al., 2014*). Usoskin et al performed an elegant single cell RNA-seq on hundreds of DRG neurons that were picked in an unbiased fashion robotically (*Usoskin et al., 2014*). We believe that our study possesses has unique features and certain advantages, as well as limitations, in relation to these studies. In our study, we performed whole population analysis of three major DRG subsets, which we followed by single cell granular profiling of hundreds of cells from the same populations. We believe advantages of beginning with a differential analysis of well-defined populations is that this facilitates correlation of the data back to function and enables a highly specific comparative analysis to be performed between major neuronal populations. Further definition of each population by shifting to a single cell strategy then allows identification of functionally defined groups of cells. The same advantages of a population based strategy is also a caveat, in that it could introduce pre-determined bias, which Usoskin et al purposely avoided by randomly picking single DRG neurons as a starting point. We note that our analysis is the only one so far to utilize parallel qRT-PCR of single cells, which we demonstrate is able to detect log-scale differences in expression (*Figure 11*), and may have better detection sensitivities than single cell RNA-seq. In a comparison of the overall datasets, we produce some similar findings with Usoskin et al, including the finding of a distinct pruriceptive population (IL31ra⁺ Group VI). However, our analysis showed higher definition of markers present in Group I and Group VII neurons, as well as Group IV neurons (which was not previously described), while Usoskin et al detected TrkB⁺ neurons whereas we did not, as these cells are not included in our sorted populations. We believe that our study and these recently published papers will be useful foundation and resource for future analysis of the molecular determinants of sensory neuron phenotype.

Somatosensory lineage neurons subserve multiple functions: nociceptive, thermoceptive, pruriceptive, proprioceptive, and tactile. It is likely that additional granular analysis at the single cell level will further refine these subsets and uncover new molecular subclasses of neurons. As genomic technologies and single cell sorting methodologies evolve current limitations (e.g., RNA quantity) will be overcome and future analysis of thousands of single cells from distinct anatomical locations, developmental time-points, or following injury/inflammation will begin to reveal even more critical information about the somatosensory system.

This transcriptional analysis illustrates an unsuspected degree of molecular complexity of primary sensory neurons within the somatosensory nervous system. Functional studies are now needed to analyze the roles of the many newly identified sensory genes in neuronal specification and action. As we begin to explore the function, connectivity and plasticity of the nervous system we need to recognize this needs a much more granular analysis of molecular identity, since even the presumed functionally relatively simple primary sensory neuron, is extraordinarily complex and diverse.

## Materials and methods

### Mice

Parvalbumin-Cre (*Hippenmeyer et al., 2005*), ai14 Rosa26TdTomato mice (*Madisen et al., 2010*) were purchased from Jackson Labs (Bar Harbor, ME) and bred in the animal facility at Boston Children's Hospital. SNS-Cre transgenic mice (*Agarwal et al., 2004*) were from Rohini Kuner (University of Heidelberg, Germany). All animal experiments were conducted according to institutional animal care and safety guidelines at Boston Children's Hospital and at Harvard Medical School.

### Ethics statement

All studies were conducted under strict review and guidelines according to the Institutional Animal Care and Use Committee (IACUC) at Boston Children's Hospital, which meets the veterinary standards

set by the American Association for Laboratory Animal Science (AALAS). The experiments were reviewed and approved by the IACUC at Boston Children's Hospital under animal protocol number 13-01-2342R.

## Immunostaining and microscopy

Mice were transcardially perfused with PBS followed by 4% Paraformaldehyde/PBS (Sigma–Aldrich, St. Louis, MO). DRG, spinal cord, plantar tissue were dissected, post-fixed for 2 hr, cryoprotected in 30% sucrose/PBS, and frozen at −80°C in Optimal cutting temperature compound (OCT, Electron Microscopy Sciences, Hatfield, PA). For plantar skin, 50 µm cyrosections were cut onto Superfrost Plus slides (Thermo Fisher Scientific, Waltham, MA) and imaged by confocal microscopy using 1 µm Z-stacks. For DRG, 14 µm cryosections were cut onto Superfrost Plus slides, stained with rabbit anti-CGRP (EMD Millipore, Billerica, MA, PC205L, 1:500), rabbit anti-tyrosine hydroxylase (EMD Milllipore, AB152, 1:1000), rabbit anti-NeuN (Millipore, A60, 1:1000), rabbit anti-Parvalbumin (Swant, Switzerland; PV25, 1:1000), followed by Alexa 488 or Alexa 647 goat anti-rabbit IgG (Life Technologies, Grand Island, NY; 1:1000) or chicken anti-neurofilament 200 (EMD Millipore, AB5539, 1:500), followed by Alexa 488 or Alexa 647 anti-chicken IgG (Life Technologies, 1:1000). For spinal cord, 20 µm cryosections were cut onto Superfrost Plus slides, stained with rabbit anti-CGRP or anti-PKCγ (1:1000), followed by Alexa 488 or Alexa 647 goat anti-rabbit IgG (Life Technologies, 1:1000). Isolectin B4-FITC (Vector Labs, Burlingame, CA; 1:1000) or Isolectin B4-Alexa 647 (Life Technologies, 1:1000) were also used for staining. Sections were mounted in Prolong Gold antifade reagent (Life Technologies) prior to imaging using an Eclipse 50i epifluorescence microscope (Nikon, Melville, NY, USA). Fluorescent DRG images were thresholded and analyzed for cell size by NIH ImageJ software. For quantification, at least eight distinct 10× fields of lumbar DRG staining from n = 3 animals were analyzed for co-localization and neuronal proportions. Statistical analysis and graphs were generated using Prism software (Graphpad, La Jolla, CA).

For whole mount imaging, lumbar dorsal root ganglia, trigeminal ganglia, sciatic nerve, plantar skin, abdominal walls were dissected and mounted in PBS under glass coverslips. Confocal microscopy was conducted using a LSM700 laser-scanning confocal microscope (Carl Zeiss, Germany), using a 10× Zeiss EC plan-NEOFLUAR dry and a 40× Zeiss plan-APOCHROMAT oil objectives, with Z-stacks imaged at 1 µm steps, collected of up to 200 µm total. Three dimensional reconstructions were rendered as maximum projection images using Volocity software (Perkin Elmer, Waltham, MA).

## RNA in situ *hybridization*

For in situ *hybridization* (ISH), mice were euthanized with $CO_2$. Lumbar L4–L6 DRGs were dissected and immediately frozen in OCT on dry ice. Tissue was cryosectioned (10–12 µm), mounted onto Superfrost Plus slides (VWR, Radnor, PA), frozen at −80°C. Digoxigenin- and fluorescein-labeled anti-sense cRNA probes matching coding (Gprc5b, Lpar3, TdTomato, Ntrk2 [Trkb], Prkcq, Nppb, Il31ra) or untranslated regions were synthesized, hybridized to sections, and visualized as previously described (*Liberles and Buck, 2006*), with minor modifications in amplification strategy. Following overnight hybridization, slides were incubated with peroxidase conjugated anti-digoxigenin antibody (Roche Applied Sciences, Indianapolis, IN, USA; 1:200) and alkaline phosphatase conjugated anti-fluorescein antibody (Roche Applied Sciences, 1:200) for 1 hr at room temperature. Tissues were washed and incubated in TSA-PLUS-Cy5 (Perkin Elmer) followed by HNPP (Roche Applied Sciences) according to manufacturer's instructions. Epifluorescence images were captured with a Leica TCS SP5 II microscope (Leica microsystems, Buffalo Grove, IL). Sequences of primers used for probe generation are listed in *Table 3*.

## Neuronal cultures and electrophysiology

For electrophysiological analysis of Parv-Cre/TdTomato and SNS-Cre/TdTomato neurons, DRGs were dissected, placed in HBSS, incubated for 90 min with 5 mg/ml collagenase, 1 mg/ml dispase II at 37°C. Cells were triturated in the presence of DNase I inhibitor, centrifuged through 10% BSA, resuspended in 1 ml of neurobasal medium, 10 µM Ara-C (Sigma-Adrich), 50 ng/ml NGF, 2 ng/ml GDNF (Life Technologies), and plated onto 35-mm tissue culture dishes coated with 5 mg/ml laminin. Cultures were incubated at 37°C under 5% $CO_2$. Recordings were made at room temperature within 24 hr of plating. Whole-cell recordings were made with an Axopatch 200A amplifier (Molecular Devices, Sunnyvale, CA) and patch pipettes with resistances of 2–3 MΩ. The pipette capacitance was decreased by wrapping the shank with parafilm and compensated using the amplifier circuitry. Pipette solution was 5 mM NaCl, 140 mM KCl, 0.5 mM $CaCl_2$, 2 mM $MgCl_2$, 5 mM EGTA, 10 mM HEPES, and 3 mM

Na$_2$ATP, pH 7.2, adjusted with NaOH. The external solution was 140 mM NaCl, 5 mM KCl, 2 mM CaCl$_2$, 2 mM MgCl$_2$, 10 mM HEPES, and 10 mM D-glucose, pH 7.4, adjusted with NaOH (Sigma-Aldrich). Current clamp recordings were made with the fast current-clamp mode. Command protocols were generated and data digitized with a Digidata 1440A A/D interface with pCLAMP10 software. Action potentials (AP) were evoked by 5 ms depolarizing current pulses. AP half width was measured at half-maximal amplitude. 500 nM Tetrodotoxin (TTX) were applied to block TTX-sensitive Na$^+$ currents.

## Flow cytometry of neurons

DRGs from cervical (C1–C8), thoracic (T1–T13), and lumbar (L1–L6) segments were pooled from different fluorescent mouse strains, consisting of 7–20 week age-matched male and female adult mice (see *Table 1*). DRGs were dissected, digested in 1 mg/ml Collagenase A/2.4 U/ml Dispase II (enzymes from Roche), dissolved in HEPES buffered saline (Sigma-Aldrich) for 70 min at 37°C. Following digestion, cells were washed into HBSS containing 0.5% Bovine serum albumin (BSA, Sigma-Aldrich), filtered through a 70 µm strainer, resuspended in HBSS/0.5% BSA, and subjected to flow cytometry. Cells were run through a 100 µm nozzle at low pressure (20 p.s.i.) on a BD FACS Aria II machine (Becton Dickinson, Franklin Lakes, NJ, USA). A neural density filter (2.0 setting) was used to allow visualization of large cells. Note: Initial trials using traditional gating strategies (e.g., cell size, doublet discrimination, and scatter properties) did not eliminate non-neuronal cells. An important aspect of isolating pure neurons was based on the significantly higher fluorescence of the Rosa26-TdTomato reporter in somata compared to axonal debris, allowing accurate gating for cell bodies and purer neuronal signatures. For microarrays, fluorescent neuronal subsets were FACS purified directly into Qiazol (Qiagen, Venlo, Netherlands). To minimize technical variability, SNS-Cre/TdTomato (total, IB4$^+$, IB4$^-$) and Parv-Cre/TdTomato neurons were sorted on the same days. FACS data was analyzed using FlowJo software (TreeStar, Ashland, OR, USA). For Fluidigm analysis, single cells or multiple cell groups from different neuronal populations were FACS sorted into individual wells of a 96-well PCR plate containing pre RNA-amplification mixtures. For microscopy, fluorescent neurons or axons were FACS purified into Neurobasal + B27 supplement + 50 ng/ml NGF, plated in poly-d-lysine/laminin-coated 8-well chamber slides (Life Technologies) and imaged immediately or 24 hr later by Eclipse 50i microscope (Nikon). Flow cytometry was performed in the IDDRC Stem Cell Core Facility at Boston Children's Hospital.

## Single neuron analysis

Flow cytometry was used to purify 100 cell groups, 10 cell groups, or single cells into 96-well plates containing 9 µl of a pre-amplification containing reaction mix from the CellsDirect One-Step qRT-PCR Kit (Life Technologies) mixture with pooled Taqman assays (purchased as optimized designs from Life Technologies). Superscript III RT Taq mix (Life Technologies) was used for 14 cycles to pre-amplify specific transcripts. We found that not every FACS sorted-well contained a cell; thus, a pre-screening method was utilized, where 2 µl from each well was subjected to two-step quantitative PCR (qPCR) for Actb (β-Actin) using fast SYBR green master mix (Life Technologies) on an Applied Biosystems 7500 machine (Applied Biosystems, Waltham, MA) using the following primers: 5'-acactgtgcccatctacgag-3' and 5'-gctgtggtggtgaagctgta-3'. Wells showing Actb Ct values <20 were picked for subsequent analysis. Using the Biomark Fluidigm microfluidic multiplex qRT-PCR platform, pre-amplified well products were run on 96.96 dynamic arrays (Fluidigm, San Francisco, CA) and assayed against 81 Taqman assays (Life Technologies). Specific assays were chosen based on differential expression by microarray analysis, functional category, and housekeeping genes (*Table 2*). Ct values were measured by Biomark software, relative transcript levels determined by $2^{-\Delta Ct}$ normalization to Gapdh or Actb transcript levels. For each transcript, outliers of 5 standard deviations from the mean were excluded (set to 0) from our analysis. A total of 334 single cells were analyzed, consisting of IB4$^+$SNS-Cre/TdT$^+$ (n = 132), IB4$^-$SNS-Cre/TdT$^+$ (n = 110), Parv-Cre/TdT$^+$ (n = 92) neurons. Spearman rank average-linkage clustering was performed with the Hierarchical Clustering module from the GenePattern genomic analysis platform and visualized using the Hierarchical ClusteringViewer module of GenePattern (MIT Broad Institute). A specific level of hierarchical clustering was used to ascertain clustered neuron subgroups. The Population PCA tool was used for principal components analysis—http://cbdm.hms.harvard.edu/LabMembersPges/SD.html. Pearson correlation analysis of specific transcripts to all 80 probes across the single cell expression dataset was generated using nearest neighbor analysis by the GenePattern platform. Histogram plots of single cell data were generated in Excel (Microsoft, Redmond, WA, USA). Dot plots showing single cell transcript data across subgroups was generated in Prism software (Graphpad).

## Statistical analysis

Sample sizes for experiments were chosen according to standard practice in the field. '*n*' represents the number of mice, samples, or cells used in each group. Bar and line graphs are plotted as mean ± standard error of the mean (s.e.m.). Data meet the assumptions of specific statistical tests chosen, including normality for parametric or non-parametric tests. Statistical analysis of electrophysiology, neuronal cell counts, and flow cytometry were by One-way ANOVA with Tukey's post-test or by unpaired, Student's *t* test. Data was plotted using Prism software (Graphpad).

## RNA processing, microarray hybridization and bioinformatics analysis

RNA was extracted by sequential Qiazol extraction and purification through the RNeasy micro kit with on column genomic DNA digestion according to manufacturer's instructions (Qiagen). RNA quality was determined by Agilent 2100 Bioanalyzer using the Pico Chip (Agilent, Santa Clara, CA, USA). Samples with RIN >7 were used for analysis. RNA was amplified into cDNA using the Ambion WT expression kit for Whole Transcript Expression Arrays (Life Technologies), with Poly-A controls from the Affymetrix Genechip Eukaryotic Poly-A RNA control kit (Affymetrix, Santa Clara, CA, USA). The Affymetrix Genechip WT Terminal labeling kit was used for fragmentation and biotin labeling. Affymetrix GeneChip Hybridization control kit and the Affymetrix GeneChip Hybridization, wash, stain kit was used to hybridize samples to Affymetrix Mouse Gene ST 1.0 GeneChips, fluidics performed on the Affymetrix Genechip Fluidics Station 450, and scanned using Affymetrix Genechip Scanner 7G (Affymetrix). Microarray work was conducted at the Boston Children's Hospital IDDRC Molecular Genetics Core. For Bioinformatics analysis, Affymetrix CEL files were normalized using the Robust Multi-array Average (RMA) algorithm with quantile normalization, background correction, and median scaling. Hierarchical clustering and principal-component analysis (PCA) was conducted on datasets filtered for mean expression values greater than 100 in any population (*Mingueneau et al., 2013*), with elimination of noisy transcripts with an intra-population coefficient of variation (CoV) <0.65. Spearman-rank average linkage analysis was conducted on the top 15% most variable probes across subsets (2735 transcripts) using the Hierarchical Clustering module, and heat-maps generated using the Hierarchical ClusteringViewer module of the GenePattern analysis platform (Broad Institute, MIT). The Population PCA tool was used (http://cbdm.hms.harvard.edu/LabMembersPges/SD.html). For pathway enrichment analysis, pairwise comparisons of specific neuronal datasets (e.g., Parv-Cre/TdTomato vs SNS-Cre/TdTomato) were conducted. Differentially expressed transcripts (twofold, p < 0.05) were analyzed using Database for Annotation, Visualization and Integrated Discovery (DAVID) (http://david.abcc.ncifcrf.gov). Pathway enrichment p-values for GO Terms (Biological Processes) or Kyoto Encyclopedia of Genes and Genomes (KEGG) pathways were plotted as heat-maps using the HeatmapViewer module of GenePattern. Differentially expressed transcripts were illustrated using volcano plots, generated by plotting fold-change differences against comparison p-values or −log (p-values). Transcripts showing low intragroup variability (CoV < 0.65) were included in this differential expression analysis. Specific gene families, including ion channels (calcium, sodium, potassium, chloride, ligand-gated, TRP and HCN channels), GPCRs and transcription factors were highlighted on volcano plots.

## Data Deposition

All microarray datasets are deposited at the NCBI GEO database (http://www.ncbi.nlm.nih.gov/) under accession number GSE55114. Data in *Supplementary files 1 and 2* are deposited at Dryad (http://dx.doi.org/10.5061/dryad.dk68t).

## Acknowledgements

We thank Olesegun Babanyi, Ta-wei Lin, Catherine Ward, Richard Bennett, Kristen Cabal, and Noreen Francis for technical support; Sriya Muraldiharan and Amanda Strominger for immunostaining and neuron quantification; Mark Hoon for Nppb probe information; Christian Von Hehn for helpful discussions on neuronal purification; Bruce P Bean, Vijay Kuchroo for helpful advice. This work was supported by CJW NIH R37 NS039518; R01 NS038253; 1PO1 NS072040-01; and the Dr. Miriam and Sheldon G. Adelson Medical Foundation. IMC received fellowship support from NIH F32 NS076297-01. Gene expression analysis were performed in the IDDRC Molecular Genetics Core facility at Boston Children's Hospital, supported by National Institutes of Health award NIH-P50-NS40828. Flow cytometry was performed in the IDDRC Stem Cell Core Facility at Boston Children's Hospital, supported by

NIH-P30-HD18655. Microarray work was conducted at the Boston Children's Hospital IDDRC Molecular Genetics Core, supported by NIH-P30-HD 18655.

## Additional information

### Funding

| Funder | Grant reference number | Author |
|---|---|---|
| National Institutes of Health | R37 NS039518; R01 NS038253; 1PO1 NS072040-01 | Clifford J Woolf |
| Dr. Miriam and Sheldon G. Adelson Medical Research Foundation | | Clifford J Woolf |

The funders had no role in study design, data collection and interpretation, or the decision to submit the work for publication.

### Author contributions

IMC, Envisioned and carried out the project, including experimental design, flow cytometry, dissections, whole population and single cell profiling, computational biology. Wrote and revised the manuscript, Conception and design, Acquisition of data, Analysis and interpretation of data, Drafting or revising the article, Contributed unpublished essential data or reagents; LBB, Author contributed to setup of microarray work, fluorescence image analysis, and RNA in situ hybridization, Acquisition of data, Analysis and interpretation of data; EKW, DES, Performed key RNA in situ hybridization experiments, microscopy, and edits/revisions of the article, Acquisition of data, Analysis and interpretation of data, Drafting or revising the article; SL, Made important contributions to electrophysiological studies and editing/revision of article, Acquisition of data, Analysis and interpretation of data; ADW, Helped with electrophysiological analysis and revising/editing of manuscript, Acquisition of data, Drafting or revising the article; SL, Performed RNA in situ hybridization experimets and data collection, Acquisition of data; GSB, Contributed to fluorescence staining and analysis of neuronal subsets, Acquisition of data, Analysis and interpretation of data; DPR, Performed behavioral analysis and revision of manuscript, Acquisition of data, Drafting or revising the article; NG, Helped with conception/design of project, and drafting/revising article, Conception and design, Drafting or revising the article; CP, Performed RNA in situ hybridization and analysis, Acquisition of data, Analysis and interpretation of data; EA, Performed technical work, fluorescence staining and analysis of neuronal subsets, Acquisition of data, Analysis and interpretation of data; VW, Performed technical work, fluorescence staining and analysis of neuronal subsets, Acquisition of data; EJC, Helped with behavioral analysis (not in manuscript) and edits/revision of manuscript, Acquisition of data, Drafting or revising the article; CLS, Contributed help with electrophysiology and revising of the article, Analysis and interpretation of data, Drafting or revising the article; QM, Analysis and interpretation of data, Drafting or revising the article; SDL, Contributed significant intellectual and experimental resources for carrying out RNA in situ hybridization work and editing of manuscript, Analysis and interpretation of data, Drafting or revising the article, Contributed unpublished essential data or reagents; CJW, Oversaw the conception and design of the project, provided overall resources, coordinated the experimental and analytical work, and was major editor of manuscript, Conception and design, Analysis and interpretation of data, Drafting or revising the article

### Ethics

Animal experimentation: All animal experiments were conducted according to institutional animal care and safety guidelines at Boston Children's Hospital, in strict accordance with the recommendations in the Guide for the Care and Use of Laboratory Animals of the National Institutes of Health. All animal work was conducted under strict review and guidelines according to the Institutional Animal Care and Use Committee (IACUC) at Boston Children's Hospital, which meets the veterinary standards set by the American Association for Laboratory Animal Science (AALAS). The experiments were reviewed and approved by the IACUC at Boston Children's Hospital under animal protocol number 13-01-2342R.

## Additional files

### Supplementary files

• Supplementary file 1. Comparison of SNS-Cre/TdT vs Parv-Cre/TdT neuron expression profiles. Differential expression analysis of microarray data from SNS-Cre/TdTomato+ neurons (n = 4) vs Parv-Cre/TdTomato+ neurons (n = 4). Transcripts are ranked by fold-change, with the following information given: Affymetrix ID, genebank accession number, gene symbol, description, average RMA normalized levels, standard deviation, fold-change, p-value and FDR.

• Supplementary file 2. Comparison of IB4 positive vs IB4 negative SNS-Cre/TdT neuron profiles. Differential expression analysis of microarray data from IB4+SNS-Cre/TdTomato+ neurons (n = 3) vs IB4−SNS-Cre/TdTomato+ neurons (n = 3). These cells were sorted from the same animals. Transcripts are ranked by fold-change, with the following information given: Affymetrix ID, genebank accession number, gene symbol, description, average RMA normalized levels, standard deviation, fold-change, p-value and FDR.

### Major datasets

The following datasets were generated:

| Author(s) | Year | Dataset title | Dataset ID and/or URL | Database, license, and accessibility information |
|---|---|---|---|---|
| Chiu IM, Woolf CJ | 2014–2015 | Whole transcriptome analysis of FACS purified somatosensory neuron subtypes and whole dorsal root ganglia tissue | GSE55114; http://www.ncbi.nlm.nih.gov/geo/query/acc.cgi?acc=GSE55114 | Publicly available at NCBI Gene Expression Omnibus. |
| Chiu IM, Barrett LB, Williams E, Strochlic DE, Lee S, Weyer AD, Lou S, Bryman G, Roberson DP, Ghasemlou N, Piccoli C, Ahat E, Wang V, Cobos EJ, Stucky CL, Ma Q, Liberles SD, Woolf CJ | 2014 | Data from: Transcriptional profiling at whole population and single cell levels reveals somatosensory neuron molecular diversity | http://dx.doi.org/10.5061/dryad.dk68t | Available at Dryad Digital Repository under a CC0 Public Domain Dedication. |

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
