## [Decision Letter]

Thank you for sending your work entitled “Transcriptional profiling at whole
population and single cell levels reveals somatosensory neuron molecular
diversity” for consideration at *eLife*. Your article has been
favorably evaluated by a Senior editor, Jeremy Nathans (Reviewing editor), and 3
reviewers, one of whom, Lee Henry, has agreed to share his identity.

The Reviewing editor and the other reviewers discussed their comments before we reached
this decision, and the Reviewing editor has assembled the following comments to help you
prepare a revised submission.

All of the reviewers were impressed with the thoroughness of your experiments and the
accompanying analysis. We believe that this data set will be of substantial interest to
neuroscientists working on DRG development and function, and that it will also be of
interest to a broad community of scientists who are working with single cell
transcriptome approaches.

Four major points to address are listed below:

1) One issue with the data is that the expression pattern of Kcnq4 seems inconsistent
between microarray and qRT-PCR. Kcnq4 has previously been found to be expressed in
low-threshold mechanoreceptors, but not nociceptors or proprioceptors (Heidenreich, et
al., 2012). In the population RNA seq data presented in the Excel files, Kcnq4 is shown
to be highly expressed in SNS-Cre+ neurons, but not detectable in Parv-Cre+
neurons, which is inconsistent with the previous publication. However, in Figure 12, Kcnq4 is identified as a transcript
specifically expressed in Group VII, which contains the Parv-Cre+ population. The
authors should address why this transcript is enriched in one population by microarray,
but another by quantitative RT-PCR.

2) In Figure 3—figure supplement 2 the
authors present their data as 'transcript fold differences'. This is a valid
means of comparing multiple samples, however, it doesn't speak to the purity of the
samples being compared. For example, if the profiled cell populations are enriched
rather than highly pure, you would expect to recover differences in any pairwise
comparison, however, the magnitude of those differences would be depressed if a
sufficient number of non-target cells contribute to the individual profiles. It is clear
that the experiments are highly reproducible (Figure 3—figure supplement 2), however, it would be useful if the authors
could show in a more absolute (not comparative) manner that their samples do not express
genes that should not be in these cell types.

3) The data presented in Figure 12 is
impressive, however, a common problem associated with single cell studies is
amplification-based noise. The authors could address this issue by showing the
expression of the 'ubiquitous' markers that they describe in the text (Scn10a,
Scn11a, Kcnc1, Kcnv1, Trpv1 and Trpa1). This could be done in a second part of Figure 12 or in the supplement to the figure.

4) The authors should mention possible limitations to their analysis, since I assume the
profiles were obtained from a single (postnatal?) time point and from sensory neurons
taken from a single (lumbar?) level. The authors should clarify at the beginning of the
results the age and rostrocaudal position of the DRG being analyzed. Genes relevant to
establishing proper circuitry may include those that are only transiently expressed
during embryogenesis, are expressed on only at certain levels (as a consequence of
intrinsic positional identities, or target dependent gene regulation).

---

## [Author Response]

*1) One issue with the data is that the expression pattern of Kcnq4 seems
inconsistent between microarray and qRT-PCR. Kcnq4 has previously been found to be
expressed in low-threshold mechanoreceptors, but not nociceptors or proprioceptors
(Heidenreich, et al., 2012). In the population RNA seq data presented in the Excel
files, Kcnq4 is shown to be highly expressed in SNS-Cre+ neurons, but not
detectable in Parv-Cre+ neurons, which is inconsistent with the previous
publication. However, in*
Figure 12*, Kcnq4 is
identified as a transcript specifically expressed in Group VII, which contains the
Parv-Cre+ population. The authors should address why this transcript is enriched
in one population by microarray, but another by quantitative RT-PCR*.

We thank the reviewers for reviewing our expression data in detail and finding that
Kcnq4 showed a mismatched expression pattern between the whole population microarray
data and the single cell qRT-PCR datasets. In reanalyzing our qRT-PCR dataset, we found
that we had made an error in labeling gene names for neuronal Group VII, where
“Spp1” was incorrectly labeled as “Kcnq4”, which is the
underlying reason for this discrepancy. We apologize for this error and we are grateful
that reviewers spotted it.

We have now reanalyzed all single cell neuronal expression data to ensure accuracy of
all the subgroup-enriched markers. In the newly added Figure 12—figure supplement 2, we show plots of all characteristic
markers of the two neuronal subgroups that have the most significantly enriched
transcripts in our dataset, Group I (Grik1, Scn11a, Mrgprd, Agtr1a, St6gal2, Pde11a,
Ggta1, Prkcq, A3galt2, Ptgdr, P2rx3, Lpar5, Mmp25, Lpar3, Casz1, Slc16a12, Lypd1, Moxd1,
Wnt2b) and Group VII (Pvalb, Car2, Spp1, Ndst4, Etv1, Gprc5b, Ano1, Pth1r, Runx3, Kcnc1,
Wnt7a, Cdh12). These transcripts are presented in order from highest to lowest
expression levels normalized to Gapdh. In this re-analysis, we found that Group I
enriched transcripts Pde11a and Lypd1 were not originally reported in our manuscript. We
have now made changes to Figure 12 and added
supplemental Figure 1 to correct this. For gene
expression markers described for Groups II, IV, V, and VI, underlying plots of gene
expression are shown in Figure 14, and
supplements 1, 2.

As the reviewers point out, our Kcnq4 population-level data and the published literature
(*Heidenreich et al, 2012*) show enrichment in
IB4^-^SNS-Cre^+^ neurons (Figure 6, Tables 2, 6). Our single cell qRT-PCR data for Kcnq4, however, is
inconclusive (Figure 12—figure supplement 2). This may be due to the Taqman probeset used, and we thus do not make
further conclusions about its single cell expression. By contrast, Spp1, which we had
originally mislabeled as Kcnq4, is definitively enriched in Group VII at high levels
compared to other neuronal groups (Figure 12—figure supplement 2).

The following specific corrections have been made to the manuscript:
“Kcnq4” has been relabeled to “Spp1” in Figure 12 and in the manuscript text. Pde11a and Lypd1 are now
added as Group I characteristics in Figure 12.

*2) In*
Figure 3—figure supplement 2
*the authors present their data as 'transcript fold differences'. This
is a valid means of comparing multiple samples, however, it doesn't speak to the
purity of the samples being compared. For example, if the profiled cell populations
are enriched rather than highly pure, you would expect to recover differences in any
pairwise comparison, however, the magnitude of those differences would be depressed
if a sufficient number of non-target cells contribute to the individual profiles. It
is clear that the experiments are highly reproducible (figure supplement 2), however,
it would be useful if the authors could show in a more absolute (not comparative)
manner that their samples do not express genes that should not be in these cell
types*.

We appreciate the point that relative fold-differences does not demonstrate purity as
well as absolute values. We have now re-plotted all myelin associated transcripts,
nociceptor associated transcripts, and proprioceptor associated transcripts as absolute
robust multi-array average (RMA) normalized expression levels (new Figure 3—figure supplement 2). These data show significant
decreases in myelin associated transcript expression in FACS purified samples relative
to whole DRG samples (p<0.001 by One-way ANOVA) and increases in neuronal
transcript expression levels (p<0.001 by One-way ANOVA). Overall signal intensity
was higher for some microarray probesets, which may reflect greater probeset sensitivity
or nonspecific background. For example, low level expression of certain transcripts were
found in sorted samples (Mpz, Mbp), while other transcripts were not detected at all
(Pmp2, Mag). Based on these data, our sorting methodology strongly enriches for neurons
compared to whole DRG tissue, but we cannot rule out the presence of small numbers of
non-neuronal cells. Thus, these data confirm the advantage of FACS sorting of neuronal
subsets vs. whole DRG tissue analysis.

*3) The data presented in*
Figure 12
*is impressive, however, a common problem associated with single cell studies is
amplification-based noise. The authors could address this issue by showing the
expression of the 'ubiquitous' markers that they describe in the text
(Scn10a, Scn11a, Kcnc1, Kcnv1, Trpv1 and Trpa1). This could be done in a second part
of*
Figure 12
*or in the supplement to the figure*.

We thank the reviewers for pointing out potential issues with amplification-based noise.
We now show the presence of these neuronal markers, as the reviewer requests (new Figure 12—figure supplement 1).
Specifically, we show the expression patterns of the ligand-gated ion channels (Trpv1,
Trpa1), voltage-gated ion channels (Scn10a, Scn11a), Kcnc1, Kcnv1), and Pvalb
(proprioceptive neuron marker) across the 334 single cell dataset. Single cell samples
show expression of at least one of these transcripts in each of the cells, indicating
their neuronal nature, as these ion channels or markers are strongly associated with
neuron-specific functions. These data also demonstrates that amplification-based noise
does not obscure the specific enrichment of specific neuronal transcripts in particular
subsets as defined by Fluidigm qRT-PCR.

*4) The authors should mention possible limitations to their analysis, since I
assume the profiles were obtained from a single (postnatal?) time point and from
sensory neurons taken from a single (lumbar?) level. The authors should clarify at
the beginning of the results the age and rostrocaudal position of the DRG being
analyzed. Genes relevant to establishing proper circuitry may include those that are
only transiently expressed during embryogenesis, are expressed on only at certain
levels (as a consequence of intrinsic positional identities, or target dependent gene
regulation)*.

We have now added the requested clarification of our analysis at the beginning of the
Results section on FACS purification and transcriptional profiling analysis. The details
of these methodologies are also in the Method section under “Flow Cytometry of
Neurons”. The limitations of our study are also discussed in the new manuscript
in relation to other studies in the Discussion.

Our population level and single cell sorting was conducted postnatally on dorsal root
ganglion neurons isolated from 7-20 week old mice (both males and females). For
microarray whole transcriptome analysis, 100 ng RNA was required for reverse
transcription, cDNA preparation using the Ambion WT Expression kit, which is a standard
protocol in Genechip hybridization facilities. We actively chose to avoid performing
pre-amplification steps using kits prior to this step, which often introduces transcript
signal biases. As a result, sufficient RNA was more difficult to obtain from sorted
populations, in particular Parv-Cre/TdTomato^+^ neurons, which make up
only 12.5±1.7% of DRG neurons). Thus, in order to obtain sufficient RNA for
analysis without introducing amplification biases, we pooled cervical (C1-C8), thoracic
(T1-T13), and lumbar DRGs (L1-L6). This is a limitation of our study, as certain
anatomical regions are potentially more enriched for particular transcripts (e.g. lumbar
vs. certical). To minimize technical variability, SNS-Cre/TdTomato (total,
IB4^+^, IB4^-^) and Parv-Cre/TdTomato neurons were sorted on
the same day. As the reviewer points out, developmental timepoint certainly influences
gene expression, as distinct embryonic stage-dependent expression patterns likely
determine specification of each subset. Preliminary analysis of neonatal
SNS-Cre/TdTomato^+^ neurons (P2) show significant differences compared
to our adult purified neurons (Data not shown). We discuss these imitations in the
Discussion, and point out that more specific and targeted analysis will be important
future studies.

While preparing the revisions to this manuscript, three papers performing expression
profiling of postnatal adult sensory neurons were published: one of which was mentioned
by Reviewer #1 (*Goswani et al,* 2014;[51]*; Usokin et al,
2014*). We have added a new paragraph in our manuscript discussing how our
study’s approach and findings relate to these papers. In particular comparison to
*Usokin et al*, we have some similar findings, including our Group VI
pruriceptive neurons (IL31ra^+^). However, our analysis showed higher
definition of Group I and Group VII neurons (distinctly more markers), as well as Group
IV neurons which they did not detect. We note that each of these studies utilized
distinct methodologies: two performed analysis of one neuronal population each
(*Goswani et al, 2014;*
[51]), and one
performed single cell RNA-seq but not population analysis (*Usokin et al,
2014*). We believe that our study possesses certain advantages as well as
limitations in relation to these studies but remains a unique and important
contribution.

In brief summary: a) *Goswami et al* profiled Trpv1 lineage neurons
compared to Trpv1-DTA whole DRG tissues. b) *Thakur et al* performed
magnetic bead selection to remove DRG non-neuronal cells, leaving neurons for a
population RNA-seq analysis. c) *Usokin et al* performed single cell
RNA-seq on hundreds of DRG neurons that were robotically picked in an unbiased fashion.
In our study, we performed whole population analysis of 3 major DRG subsets, which we
followed by single cell granular profiling of hundreds of cells from the same
populations. We believe one advantage of our study is beginning with differential
analysis of well-defined and functionally relevant populations. The two published
population level studies did not perform comparative analysis with other distinct
populations. We then further define each of our populations in detail using the single
cell strategy, which gives us high-resolution analysis of functionally defined groups of
cells. The same advantages of a population based strategy is also a caveat, in that it
could introduce pre-determined bias, which *Usokin et al* purposely
avoids by randomly picking single DRG neurons as a starting point. We note that our
analysis is the only one to utilize parallel qRT-PCR of single cells, which we
demonstrate is able to detect log-scale differences in expression (Figure 11), and we believe may have better detection sensitivities
compared to single cell RNA-seq. As genomic technologies and single cell sorting
methodologies evolve, the limitations (e.g. RNA quantity) will be overcome and future
analysis of thousands of single cells from distinct anatomical locations, developmental
time-points, or following injury/inflammation will begin to reveal even more critical
information about the somatosensory system. We believe that our study and these recently
published papers are a critical foundation for such future analysis.